# Contrasted geomorphological and limnological properties of thermokarst lakes formed in buried glacier ice and ice-wedge polygon terrain

Stéphanie Coulombe[1 2 3], Daniel Fortier[2 3], Frédéric Bouchard[3 4], Michel Paquette[5],

Simon Charbonneau[2 3], Denis Lacelle[6], Isabelle Laurion[3 7], Reinhard Pienitz[3 8]

[1] Polar Knowledge Canada, Cambridge Bay, X0B 0C0, Canada
[2] Department of Geography, Université de Montréal, Montréal, H2V 2B8, Canada
[3] Centre for Northern Studies, Université Laval, Quebec City, G1V 0A6, Canada
[4] Department of Applied Geomatics, Université de Sherbrooke, Sherbrooke, J1K 2R1 Canada
[5] Ecofish Research Ltd, Squamish, V8B 0V2, Canada
[6] Department of Geography, Environment and Geomatics, University of Ottawa, Ottawa, K1N 6N5, Canada
[7] Centre Eau Terre Environnement, Institut national de la recherche scientifique, Quebec City, G1K 9A9, Canada
[8] Department of Geography, Université Laval, Quebec City, G1V 0A6, Canada

*Correspondence to*: Stephanie Coulombe (stephanie.coulombe@polar.gc.ca) and Daniel Fortier (daniel.fortier@umontreal.ca)

**Abstract.** In formerly glaciated permafrost regions, extensive areas are still underlain by a considerable amount of glacier ice buried by glacigenic sediments. It is expected that large parts of glacier ice buried in the permafrost will melt in the near future, although the intensity and timing will depend on local terrain conditions and the magnitude and rate of future climate trends in different Arctic regions. The impact of these ice bodies on landscape evolution remains uncertain since the extent and volume of undisturbed relict glacier ice are unknown. These remnants of glacier ice buried and preserved in the permafrost contribute to the high spatial variability in ground ice condition of these landscapes, leading to the formation of lakes with diverse origin, morphometric and limnological properties. This study focuses on thermokarst lake initiation and development in response to varying ground ice conditions in a glacial valley, on Bylot Island (Nunavut). We studied a lake-rich area using lake-sediment cores, detailed bathymetric data, remotely sensed data and observations of buried glacier ice exposures. Our results suggest that initiation of thermokarst lakes in the valley was either triggered from the melting of buried glacier ice or intrasedimental ice and ice wedges. Over time, all lakes enlarged through thermal and mechanical shoreline erosion, as well as vertically through thaw consolidation and subsidence. Some of them coalesced with neighbouring water bodies to develop larger lakes. These glacial thermokarst lakes formed in buried glacier ice now evolve as "classic" thermokarst lakes that expand in area and volume as a result of the melting of intrasedimental ground ice in the surrounding material and the underlying glaciofluvial and till material. It is expected that the deepening of thaw bulbs (taliks) and the enlargement of Arctic lakes in

response to global warming will reach undisturbed buried glacier ice where it is still present, which in turn will substantially alter lake bathymetry, geochemistry and greenhouse gas emissions from Arctic lowlands.

## 1 Introduction

Arctic landscapes underlain by ice-rich permafrost are highly vulnerable to climate change and permafrost degradation (Segal et al., 2016; Rudy et al., 2017; Lewkowicz and Way, 2019; Kokelj et al., 2017; Nitzbon et al., 2020; Douglas et al., 2021). These ice-rich permafrost landscapes are experiencing thermokarst, through the thawing of near-surface ice-rich permafrost and/or the melting of ice wedges or massive ice, which may result in land subsidence and ponding (Kokelj and Jorgenson, 2013; Farquharson et al., 2019; Liljedahl et al., 2016; Edwards et al., 2016; Jorgenson and Osterkamp, 2005). In flat-lying terrains, thermokarst processes often result in the formation of numerous wetlands, ponds and lakes. This creates or modifies existing 'limnoscapes' (lake-rich landscapes) through thermal and mechanical erosional processes as well as thaw consolidation and subsidence beneath waterbodies (Bouchard et al., 2020; Grosse et al., 2013; Shur et al., 2012; Plug and West, 2009). The formation and growth of these lacustrine ecosystems have important effects on the hydrology, ecology, biogeochemistry and geomorphology of affected landscapes (Vonk et al., 2015). Shoreline erosion may affect key biogeochemical processes within these lakes, such as the burial of organic matter in sediments, and its degradation and release as greenhouse gases (GHG; $CO_2$ and $CH_4$) to the atmosphere (Matveev et al., 2016; Vonk et al., 2015; Heslop et al., 2020). For example, the synthesis study by Wik et al. (2016) showed that lakes and ponds north of ~50°N are large methane emitters (notably glacial/postglacial lakes due to their larger areal extent), equivalent to roughly two-thirds of the inverse model calculation of all natural methane sources in the region. Lake basin morphology also influence GHG flux patterns during the open-water season by affecting the mixing regime (Prėskienis et al., 2021; Hughes-Allen et al., 2021).

The extent to which permafrost degradation occurs is dependent on the distribution and abundance of ground ice. In formerly glaciated permafrost regions, extensive areas still contain a considerable amount of glacial ice buried beneath glacigenic sediments (Belova, 2015; Coulombe et al., 2019; French and Harry, 1990; Ingólfsson and Lokrantz, 2003; Kanevskiy et al., 2013; Swanger, 2017; Dyke and Savelle, 2000; Lakeman and England, 2012). Remnants of buried glacier ice remain stable as long as the ground temperature is below freezing, and the active layer thickness (i.e. depth of annual thawing) does not exceed the depth to the massive ice body (Shur, 1988). Ice-cored moraines landscapes may lose their buried ice cores thousands to millions of years after the major glacial retreat (Bibby et al., 2016; Coulombe et al., 2019; Lacelle et al., 2007; Swanger, 2017). The persistence of thick beds of buried Pleistocene glacier ice in contemporary permafrost environments indicates that deglaciation is still incomplete (Astakhov and Isayeva, 1988; Everest and Bradwell, 2003; Kaplanskaya and Tarnogradskiy, 1986; Lenz et al., 2013). The broad distribution and the substantial amount of ground ice in glaciated permafrost landscapes make it highly vulnerable to disturbances, such as thermokarst, under the ongoing climate warming (Kokelj et al., 2017; Segal et al., 2016). As such, some of these landscapes are now entering a second phase of landscape evolution (Astakhov and Isayeva,

1988, Everest and Bradwell, 2003). For example, on hillslopes, the thawing of permafrost terrain underlain by remnants of glacial ice triggered mass wasting processes, such as retrogressive thaw slump and active layer detachment slides (Kokelj et al., 2017; Rudy et al., 2017). In flat or very gently sloping terrain, formation and evolution of ponds and lakes are typically associated with the melting of intrasedimental ice, such as ice wedges and segregation ice (Bouchard et al., 2017; Grosse et al., 2013). These lakes tend to be shallow, with deeper central pools (~ 2–5 m) and shallow littoral shelves (~ 1 m), or shallow flat-bottomed basins (Grosse et al., 2013; Hinkel et al., 2012; Bouchard et al., 2020; Jorgenson and Osterkamp, 2005). It is generally recognised that numerous Arctic lakes were formed during deglaciation in depressions left by in-situ melting of stagnant blocks of glacier ice (also named *kettle lakes* or *postglacial lakes*). However, very few studies have linked lake inception to the thawing of sediments containing glacier ice that had been buried and preserved in permafrost for decades to millennia (Henriksen et al., 2003; Worsley, 1999; Astakhov and Isayeva, 1988). As a result, there is little information on the spatial distribution and abundance and evolution on these glacial thermokarst lakes in modern paraglacial and periglacial environments.

The Quaternary geology of the eastern Canadian Arctic records several glaciations by ice sheets and local mountain glaciers, which means that the landscape stores vast amounts of buried glacial ice, and there is potential for significant postglacial landscape change associated with the ablation of this buried ice. The resulting landscape can be covered with a large number of thermokarst lakes of diverse origin that impact their physical and limnological properties. This study builds on the findings of Coulombe et al. (2019) conducted on Bylot Island (Nunavut), where blocks of stagnant ice became separated from an ice stream flowing from the Foxe Dome of the Laurentide Ice Sheet and subsequently buried by aggradation of glaciofluvial sands and gravels at the margins of the receding glacier. Subsequent neoglacial cooling resulted in widespread permafrost aggradation and preservation of this glacial ice. Here, we investigate the inception and evolution of twenty-one lakes from the lower reach of a glacial valley on Bylot Island, which presents heterogeneous permafrost ground ice conditions. We hypothesised that thermokarst lakes have different origins and exhibit differences in their morphological and limnological conditions as well as future sensitivity to change. In the Qarlikturvik Valley, remnants of buried glacier ice in lowlands slowly melted during the Holocene, which created deep depressions that formed *glacial thermokarst lakes*, while the thawing of an ice- and organic-rich polygonal terrace created shallow thermokarst lakes. The specific objectives were therefore (1) to compare the morphological and limnological properties of these two types of thermokarst lakes; (2) to examine the link between the spatial pattern of lakes and past glacier positions in the Qarlikturvik Valley and broader southern plain, and (3) to develop a conceptual model of lake inception and evolution, with a focus on lakes formed by the delayed melting of buried glacier ice.

## 2 Study area

The study area is in the Qarlikturvik Valley (73˚09' N, 79˚57' W) on the southwest plain of Bylot Island, in the Canadian Arctic Archipelago (Fig. 1a). The landscape was glaciated several occasions in the late Quaternary by both local mountain glaciers and the Laurentide Ice Sheet (LIS; Klassen, 1993). The study area was most likely a confluence zone between LIS ice and local alpine glaciers with glacier ice flowing out of major valleys but Laurentide ice flowing into the southern plain and up the valleys (Dyke and Hooper, 2001; Lacelle et al., 2018). The maximum extent of the LIS is outlined by the Eclipse moraine, a major moraine system across the outer coastal mountains of Bylot Island and parts of adjacent Baffin Island (Klassen and Fisher, 1988). Today, Bylot Island remains 40% glacierised as numerous valleys and piedmont glaciers still flow from the local ice cap and terminate in lowlands underlain by sedimentary rock of Cretaceous-Tertiary ages (Dowdeswell et al., 2007). The Qarlikturvik Valley is one of the many U-shaped glacial valleys with ice-rich sediments dating back to the Late Pleistocene and Holocene, which are highly susceptible to thermokarst (Fortier and Allard, 2004; Bouchard et al., 2020). The valley contains abundant and diverse water bodies, including a proglacial river, lakes, trough and polygon ponds, small streams, and thermos-erosion gullies (Godin et al., 2014; Muster et al., 2017; Prėskienis et al., 2021). With glaciers ending within the continuous permafrost zone, this lake-rich valley represents a typical glaciated valley geosystem that incorporates numerous depositional environments associated with ice-marginal, proglacial, paraglacial and periglacial processes, which makes it an ideal location to study ice types and thermokarst lake development under varying ground ice and terrain conditions. In the Qarlikturvik Valley, mounds of reworked till and ice-contact stratified sediments mark former positions of the glacier margins (Fig. 1b). The earliest postglacial radiocarbon date from marine shells retrieved from marine clays is 11,331 cal yr BP (IntCal20), suggesting that the valley was partially ice-free by this time (Allard, 1996). About 2–3 meters of ice-rich Quaternary silt and sand derived from aeolian deposition, interstratified with peat, overlies ice-poor glaciofluvial outwash deposits (Fig. 1b; Fortier and Allard, 2004). Syngenetic ice wedge growth has created extensive polygonal patterned ground. Thermokarst is an active landscape change mechanism operating in the valley, as demonstrated by the abundance of lakes, thermo-erosional gullies and thaw slumps within the study area (Bouchard et al., 2020; Fortier et al., 2007; Godin et al., 2014). Previous work in the area have examined various aspects of thermokarst lake dynamics such as GHG exchanges (Bouchard et al., 2015a; Prėskienis et al., 2021), photochemical and microbial decomposition of organic matter (Laurion et al., 2021), microbial diversity (Negandhi et al., 2014), nutrient inputs from the goose colony (Côté et al., 2010) and methylmercury (MacMillan et al., 2015), as well as lake development in syngenetic ice-wedge polygon terrain (Bouchard et al., 2020).

The mean annual air temperature at Pond Inlet for the 1981–2010 normal is −14.6°C, which is 0.5°C higher than the previous 1971–2000 record (Environment Canada, 2021). The mean annual precipitation for the 1981–2010 period was 189 mm yr$^{-1}$, with rainfall representing 91 mm. Bylot Island is located within the continuous permafrost zone. Permafrost thickness was estimated to be at least 200–400 m based on shallow ground temperature measurements on the island (Moorman, 2003). On average, the active-layer thickness varies between 0.3 and 0.7 m in peaty and silty soils, to more than 1 m in drained

unvegetated sands and gravels (Allard et al., 2020). Thawing and freezing indices averaged (1981–2010 period) 473 degree-days above 0°C and 5736 degree-days below 0°C, respectively (Environment Canada, 2021).

## 3 Materials and methods

Two spatial scales were used to investigate the role of buried glacier ice in the formation and evolution of thermokarst lakes. First, we focused on the Qarlikturvik Valley (~ 75 km$^2$), where buried Pleistocene glacier ice has been found in permafrost (Fig. 1; Coulombe et al., 2019). We examined the morphology and conducted bathymetric surveys of 21 lakes and analysed lake sediment cores from two of these lakes to infer probable lake origin. We also analysed water column profiles of temperature and dissolved oxygen of these same lakes. The studied lakes are among the largest in the valley and most of them are close to former glacier positions. Then we examined the spatial distribution of lakes on the broader coastal plain of Bylot Island (~ 122 km$^2$) to link the extension of former local and regional glaciations to lake distribution.

### 3.1 Landforms, surficial deposits and lake mapping

We used contemporary high-resolution GeoEye satellite imagery (2010, pixel = 0.5 m), WorldView-1 (2010, pixel = 0.5 m) and ArcticDEM data (pixel = 2 m) to map lakes, Quaternary surficial deposits and landforms in the Qarlikturvik Valley. We used field- and remote-based data to map glacier frontal positions of glacier C-79 and C-93 to investigate the formation of new lakes in the valley at the termini of these glaciers over the past 60 years: 1) historical aerial photos (1961, 1982; National Air Photo Library) 2) GeoEye satellite imagery (2010, pixel = 0.5 m); 3) Sentinel-2 (2016, 2020, pixel = 10 m) and 4) field measurements using a real-time kinematic (RTK) global positioning system (July 2011;Trimble R8). The positions refer to the contact between the ice and moraine material. A Sentinel-2 image mosaic (2016, pixel = 10 m) of the southern plain of Bylot Island served as the basis for mapping the water bodies outside the valley (Copernicus, 2016). We also used the Google Earth Engine Timelapse dataset (2000-2019) to visually assess terrain change and sediment accumulation at the glacier terminus based on Tasseled cap (TC) trend analysis of a Landsat image stack (Fraser et al., 2012; Gorelick et al., 2017; Nitze and Grosse, 2016). The tasseled cap transformation reduces the Landsat reflectance bands to three orthogonal indices called brightness, greenness and wetness (Crist and Cicone, 1984). Data processing and analyses were performed using QGIS (v.3.16; QGIS Development Team, 2021). To extract all water bodies, we used the reflectance properties of water in the Green and NIR bands (McFeeters, 1996). Because water bodies have high 'Normalized Difference Water Index' (NDWI) values, a simple thresholding technique was used to isolate most water features. Lake shorelines were extracted as vector data and converted to polygon topology. Lakes smaller than 1000 m$^2$ were automatically removed from the analysis to exclude polygon ponds and collapsed ice-wedge troughs filled with water.

## 3.2 Distribution of lakes in the valley and the southern plain of Bylot Island

We examined the spatial distribution of lakes to examine possible association with past glaciers positions in the Qarlikturvik Valley and the broader southern plain on the island. This can provide additional evidence on the glacial origin of lakes because these ice-marginal zones often comprise discrete bodies of glacier ice left behind by a retreating glacier and buried underneath sediment. To map the density of lakes in the Qarlikturvik Valley and the broader southern plain of Bylot Island, a kernel density estimation was performed using the 'spatstat' package in R (v. 3.5.3; Baddeley et al., 2019; R Core Team, 2021). Input for kernel density came from lake centroids obtained from the vector polygon, which were calculated automatically in R as the geometric center of the lakes. We defined the extent as all areas of the Qarlikturvik Valley and the broader southern coastal plain of Bylot Island, excluding the bedrock outcrops, slopes ($> 5°$), glaciers and outwash plains. To analyse lake spatial patterns, we also performed a clustering analysis using the inhomogeneous pairwise correlation function with 100 Monte Carlo simulations and 95% confidence interval, which accounts for spatial inhomogeneity in lake locations (quadra test; $p = 0.001$). This function considers the intensity (density) of the observed points by simulating completely spatially random point patterns based on the average intensity in the observed point pattern. This technique allows distinguishing between dispersed ($R > 1$) and clustered ($R < 1$) spatial patterns by comparing the observed point patterns against the expectation for a randomly distributed sample population (CSR model), which assumes that the objects can be distributed anywhere in the region of interest. A high spatial clustering suggests that the spatial distribution of lakes is dependent on an external variable which we interpreted as the probable presence of patches of buried glacier ice.

## 3.3 Lake morphology in the Qarlikturvik Valley

Detailed bathymetric data were collected for 21 lakes across the valley using a Humminbird 859XD Sonar with a built-in global positioning system. Lake bathymetric surveys were conducted with an inflatable boat when the lake was free of ice (June to August). Geographic location and water depth were recorded each second along transect lines that were spaced at approximately 5 to 25 m (depending on lake size) to entirely cover the lake. Some uncontrolled conditions have degraded the accuracy of the survey, such as the presence of littoral vegetation and waves, especially in shallow lakes. Depth and location data were imported into QGIS for visualization and additional processing. Initial processing included the removal of spurious data points (outliers) such as single-point depths located substantially above or below the general depth of lake-bottom. We used a spline algorithm to generate an interpolated surface from the individual depths. Ground penetrating radar (GPR) surveys with 50 Hz antenna were conducted across frozen lakes to investigate the lake-bottom morphology (see Supplementary material S1 for further details). For each lake, we also calculated the area, perimeter, elongation ratio (long axis/short axis), and shoreline development or $D_L$ from the digitised shoreline polygons to compare lake metrics and determine if they can be used to discriminate between thermokarst lakes formed in ice-wedge polygon terrain and thermokarst lakes formed by the melting of buried glacier ice. For comparison, the morphological attributes of glacial thermokarst lakes formed in proglacial outwash deposits in front on glaciers C-79, C-93 and C-67 were also calculated. Very few studies have examined glacial

thermokarst lake morphology, but studies on kettle lakes report enclosed and steep-sided depressions, roughly circular and inverse-conical (Fay, 2002; Gorokhovich et al., 2009; Borsellino et al., 2017). The shoreline development ratio ($D_L$) is a standard measure of the complexity of the shoreline, which is the ratio of the length of the shoreline of a lake (i.e. perimeter) to the circumference of a circle of area equal to that of the lake (Equation 1; Hutchinson, 1957).

$$D_L = \frac{Perimeter}{2\sqrt{Area * \pi}} \tag{1}$$

$D_L$ for a perfect circle is 1.0, and its value increases (>>1) as the shape of the lake surface deviates from that of a circle, indicating the shoreline is more dendritic or irregular. Glacial thermokarst lakes should have low complexity values (~1) whereas thermokarst lakes expanding laterally in ice-wedge polygon terrain should be more irregular and have values >1. A highly indented shoreline may also indicate coalescent lakes formed by shoreline expansion. An elongation ratio (ER) of 1 indicates a circular object with increasing *ER* values for more elongate forms. Correlation between shoreline morphology variables and basin morphometry (maximum depth) were tested using the non-parametric Kendall tau rank correlation for non-normally distributed data. All statistical tests were run in the open-source software R (R Core Team, 2021).

**3.4 Stratigraphic profiles of lake bottom sediments**

We selected two nearby lakes (IWT1 and GT1) exhibiting different morphometry to compare the stratigraphic profiles of lake bottom sediment. According to the bathymetric surveys, lake GT1 is the deepest in the valley (max. depth = 12.2 m), and it lies directly next to an ice-contact deposits mound. We also sampled lake IWT1 (max. depth = 4.1 m) as lake bottom imagery revealed submerged ice-wedge polygons (~1 m depth) and degraded ice-wedge troughs, which confirmed that this lake is evolving through the melting of permafrost intrasedimental ice and ice wedges (see video supplement in Bouchard et al., 2020). Two sediment cores of 109 cm and 114 cm were collected in spring 2015 from lakes IWT1 and GT1, respectively, through a 2-m thick ice cover using a 7-cm diameter handheld percussion corer (Aquatic Research Instruments), sealed, and returned to the laboratory where they were stored in the dark at 4˚C. Coring occasionally caused minor deformations to the sediments owing to friction and pressure along coring tubes. Both cores were observed under X-ray computed tomography (CT), allowing to visualize and reconstruct the internal structure (2D and 3D) of the cores. Details on the CT scanning procedure are provided in supplementary material S6. Facies were identified based on visual inspection and physical properties, including sedimentary structures, grain size, colour, and density. Percentage dry weight was determined for all samples (drying overnight at 105°C). Organic matter content was determined by weight loss (loss-on-ignition, LOI), following a combustion of dried samples at 550°C for 4 h (Heiri et al., 2001). Sediment grain size was measured in triplicates using a Malvern Mastersizer 2000 and Hydro2000G liquid handling unit. Bulk sediment and fossil plant fragments were radiocarbon-dated by accelerator mass spectrometry (AMS) at Keck Carbon Cycle AMS Facility (University of California, Irvine, CA, USA). Calibrated ages (cal yr BP) were calculated using "CALIB 8.2" (Stuiver et al., 2021; IntCal20 dataset, Reimer et al., 2020). In the case of lake IWT1, facies are described in more details based on other proxies, such as organic content and fossil diatoms (Bouchard et al., 2020; see Figs. 4 and 5). Finally, diatom assemblages were investigated in sediment sections from lake GT1 for comparison purposes

with those of lake IWT1, presented in Bouchard et al., (2020). Sediment samples (n=16) were prepared for diatom taxonomic identification using standard procedures in the Aquatic Paleoecology Laboratory (Laval University, Canada). The cleaned diatom samples were dried on glass cover slips and mounted in Naphrax® mounting medium. Diatoms were identified and counted at 1000X magnification using a Leica DMRB microscope.

## 3.4 Water column profiles of temperature and dissolved oxygen

We profiled the water column of lakes IWT1, GT1, and lake GT2 in late winter under the ice cover (early June 2015) and during the ice-free period (July and August 2015) to examine differences in water temperature and dissolved oxygen (DO) between lake types. Discrete profiles were measured manually from lakes IWT1 and GT1 with a ProODO profiler (YSI Inc.), while submersible temperature loggers (Vemco Minilog-II-T installed at 2, 4, 6, 8 and 10 m depth) and DO loggers (PME MiniDOT; 2 and 9 m depth) were installed in lake GT2. The loggers recorded annual cycles of stratification (1h frequency), from which profiles were selected to match the discrete profiles obtained from the other two lakes. Sensor specifications can be found in Prėskienis et al. (2021).

## 4. Results

### 4.1 Distribution of lakes in the valley and southern plain of Bylot Island

Using remote sensing classification, we detected 845 lakes larger than 1000 $m^2$ within the study area (total lake area reaching 14 $km^2$ over ~ 1700 $km^2$, or 0.8% of the area), of which 189 lakes (totalling 1.6 $km^2$) are in the Qarlikturvik Valley (~ 122 $km^2$; 1.3% of the area). The spatial distribution of the lakes showed a significant aggregation pattern in both Qarlikturvik Valley and the southern coastal plain of Bylot Island (Fig. 2). Patterns of distribution emerge in the valley with higher densities, 10 to 85 lakes per $km^2$, detected nearby mounds of ice-contact deposits or in areas of unvegetated moraine in front of glaciers C-79 and C-93 (Fig. 2a). A third group of lakes is also observed on the plateau bordering glacier C-93. According to the point pattern analyses, the lakes in the Qarlikturvik Valley show significant clustering in short distance (far above the 95% confidence envelope; r < 0.85 km) and a regular distribution further away (r > 0.85 km; Fig. S2a). On the southern plain of Bylot Island, the highest densities occur directly in front of contemporary glaciers and within the extent of local mountain glaciations and LIS, with up to 40 lakes per $km^2$ (Fig. 2b). The observed points (lake centroids) show considerable clustering at smaller distances (< 3.3 km), but show regularity beyond ~ 4 km (Fig. S2b). Because the highest densities were observed in association with past glacier and ice sheet margins (LIS), we also analysed the formation of lakes in front of glaciers C-93 and C-79 since their last major advances during the Little Ice Age (LIA; 120±80 [14]C yr BP; Klassen, 1993). From LIA to 2020, 383 new glacial thermokarst lakes developed as a result of glacier retreat (~2 km; Fig. 2c). In addition, the TC trend analysis revealed sediment accumulation at the front of the receding glaciers between 2000 and 2019 as represented by red colours

(drier and unvegetated areas) on TC images (Fig.2d). This shows the active burial of glacier ice at the front of glaciers C93 and C79, thereby providing a modern analogue for the past burial of ice when the glacier was several km further down-valley.

## 4.2 Lake morphology in the Qarlikturvik Valley

In the valley, we identified two groups of lakes according to their depth range and lake-floor morphometry (Fig. 3). Table 1 summarizes the characteristics of the lakes (n=21) for which bathymetric data were collected in 2015. The first group of lakes (deep; n=8) stands out by their greater depths and sizes, and in some cases, the presence of multiple sub-basins. The maximum measured depths recorded in these lakes range from 5.9 to 15.4 m. Most of these lakes are characterised by a relatively deep central lake basin surrounded by shallower areas, ranging between 0.5 and 1.0 m (mean depth= $0.6 \pm 0.4$ m; Table 1). Three lakes (GT2, GT5, GT7) have two or three steep-sided and confined sub-basins that are surrounded by a relatively shallow marginal platform. The GPR profiles indicated that these deeper lakes usually have smoother microtopography at the lake bottoms, whereas lake GT2 also exhibits an irregular lake floor micromorphology in the shallowest areas (Fig. S1), with submerged polygon-patterned ground and degraded ice wedge under frost crack troughs (Bouchard et al., 2020). The bathymetric map of lake GT2 also revealed a deeper depression that is aligned with a lakeside thaw slump exposing buried glacier ice (Figs. 3 and 4). The bottoms of lakes GT1 and GT2 are, respectively, 5.5 m and 5 m below the current sea level. In the valley (zones 1 and 2), the deeper lakes are located near mounds of stratified ice-contact glaciofluvial deposits. The second group of lakes (shallow; n=13) showed markedly different characteristics (Fig. 3). At the lake scale, these shallow water bodies have relatively flat and homogeneous beds with a deeper central basin surrounded by shallower nearshore zones (< 2 m deep). The lake floor is irregular at a finer scale (microtopography), which is attributed to submerged polygons (see the video supplement in Bouchard et al., 2020). These lakes have maximum depths ranging between 1 and 4 m, with mean depth reaching $1.4 \pm 0.7$ m across their platforms (Fig. 3). They are characterised by irregular shorelines, which generally follow the deep troughs caused by the melting of ice wedges from their tops. Despite the above-stated differences, lakes from both subgroups present similar shapes and shoreline characteristics as their morphometric properties (area, perimeter, elongation ratio, complexity) were not significantly different (Mann–Whitney–Wilcoxon test, $p > 0.05$; Fig. S3). In addition, mean or maximum depths did not show any significant correlation with the other morphometric variables ($p > 0.1$). In addition, glacial thermokarst lakes (n=490) located near the front of glaciers C-79, C-93 and C-67 have an average shoreline development index of $0.9 \pm 0.1$ and average elongation ratio of $1.7 \pm 0.5$, indicating the shorelines are relatively regular and are mostly oval-shaped.

## 4.3 Stratigraphic profiles of lake bottom sediments

Four distinct lithofacies or units, labelled from core bottom to top, were identified in lake GT1 (max. depth: 12.2 m) based on visual analysis of CT-scan images and field description: A) sandy silt and gravel with interspersed peat/organic debris (114-89 cm); B) sandy silt fibrous peat (89-80 cm); C) laminated to massive sandy silt (80-68 cm); D) sandy silt gyttja (organic lacustrine mud; 68-0 cm; Fig. 5). The lower unit (A) is a weakly stratified black and yellowish-brown fine sediments (coarse

silt and fine sand) and gravel with scattered organic material, which was dated near its top (95 cm) at 3531 cal yr BP (3330 $^{14}$C BP; Fig. 5). The coring operation did not reach the bottom of unit A, so its total thickness is unknown. Compared to other units, it has a higher mean density (2.0 to 2.5 g cm$^{-3}$), typical of dominantly mineral material. Unit B consists of dry and fibrous organic-rich material (peat) with fine sand and silt. Unit C consists of laminated to massive sandy silt containing very sparse and fine gravels, which is massive in its uppermost 5 cm. This unit also displays sharp lower and upper contacts, and includes some deformation structures, caused by the coring operation (layers bended downward near the coring tube walls). Bulk sediment near the top of this unit (73 cm) has been dated at 4036 cal yr BP (3700 $^{14}$C BP ; Fig. 5). The uppermost unit (D) is composed of laminated gyttja that grades upwards into soft and loose gyttja. The CT-scan image along with the LOI profile show that the mineral input steadily decreases towards the middle of unit B. Light-coloured thin laminae of silt (0.3 and 0.9 cm) are common in the upper part of the sequence. The upper section of this unit (50-0 cm depth) is less compact compared to deeper sediments (> 50 cm depth) and it has a high water content that becomes dryer towards the bottom of the unit (gravimetric water content decreases from about 80% at the surface to 15% at the bottom; Fig. 5). A similar trend is also observed for the organic content as it decreases from ~15% at the surface to near 1% in the lower portions of the core (Fig. 5). Among the 16 levels analysed for fossil diatoms, only 8 contained identifiable diatom taxa. These species were restricted to units A and B only, whereas upper units (C, D) contained only dispersed fragments that could not be identified.  In the 4 levels ranging from 90-90.5 to 74-74.5 cm (= levels 5 – 8), clastic debris was abundant but also diatoms, The majority of these were fragmented, but there were also a few remarkably well-preserved and intact specimens in the assemblage belonging to the genera *Eunotia* and *Cymbella*. *Eunotia* species are known to be adapted and associated with mosses (bryophytes) in peat and humid environments, sometimes exposed to the air, rich in *Sphagnum* mosses. They indicate circumneutral to acidic, oligotrophic (nutrient-poor) and shallow environments. *Cymbella* species (e.g., *C. cistula*) live as epiphytes on the stems and leaves of freshwater aquatic plants, while all other identified specimens of less abundant genera belonging to *Pinnularia* (e.g., *P. maior*, *P. balfouriana* = aerophilic species), *Caloneis*, *Navicula* (*N. pseudoscutiformis*), *Achnanthes*, *Tabellaria* and *Cocconeis* (*C. placentula* = species which attaches to rocky surfaces) reflect a cold shallow freshwater environment, circumneutral to slightly acidic, oligotrophic to ultraoligotrophic, and with a rocky/sandy substrates carpeted with mosses and a few aquatic plants. These are exclusively elongate (= pennate) and benthic diatoms living on substrates, and the absence of deepwater centric (planktonic) species reflects a very shallow water body with frequently alternating wet and dry conditions.

Three lithofacies were identified in lake IWT1 (max. depth: 4.1 m): A) organic-poor sandy silt (109-80 cm); B) organic-rich sandy silt interstratified with peat (80-10 cm); C) sandy silt gyttja (organic lacustrine mud; 10-0 cm; Fig. 5). More details on lake sediment stratigraphy of lake IWT1 (named *Gull Lake*) are available in Bouchard et al. (2020). Unit A is composed of sand and gravel with scattered, cm-scale peat and organic debris. The lower section of this unit has a higher density (~ 2.0 g cm$^{-3}$) and a low organic content (mean: 7.7% ± 3.7) compared to the upper sections of the core. The bottom deposit of unit A contains organic matter older than 4805 ± 15 $^{14}$C yr BP) (5507 cal yr BP; 1σ range: 5584–5586), based on dating of a woodfragment at a depth of 108 cm (Bouchard et al., 2020). Unit B consists of medium to dark brown peat, dated to 4070 ±

45 [14]C yr BP (35 cm; 4567cal yr BP; 1σ range: 4444–4789), and interbedded with mm- to cm-thick silt and sand laminations, and with gradational upper and lower boundaries. Throughout the unit, sand-silt layers (aeolian) are roughly interbedded with layers of organic detritus as recorded by shifts in organic matter contents. Towards the upper boundary, unit B progressively grades into dark brown gyttja (unit C). The upper part of unit C is faintly stratified, as the organic-rich material becomes

regularly interspersed with silty material. These silty laminae (0.3 to 0.9 cm) are visually distinguishable by their light-grey colour on the CT-scan image and their higher density. The top sediments have the highest water (70.6% ± 6.1) and organic matter contents (20.3% ± 5.2) compared to the bottom units. Contrary to lake GT1 the boundaries between the units are diffuse. The bottom of this unit (bulk sample collected at 10 cm) yielded an age of around 2100 ± 20 [14]C yr BP (2061 cal yr BP; 1σ range: 2004–2101). Fossil diatom assemblages in the 3 units reflect changes in the hydro-climatic conditions and available

substrates (both terrestrial and aquatic) in the past. Taxa in Unit A show a poor diversity and are generally associated with cold, organic-poor and mostly alkaline (pH ~ 8) waters, typical of Arctic streams; taxa in Unit B show much higher diversity and reflect permafrost peatland environmental conditions (i.e. shallow tundra ponds in organic-rich ice-wedge polygon terrains); and taxa in Unit C are dominated by strictly aquatic (both benthic ) species generally living in organic-rich, high-nutrient deeper waters (Bouchard et al., 2020; see Fig. 5 and accompanying section 4.2.2).

## 4.4 Water column profiles of temperature and dissolved oxygen

Profiles done in early June under the ice cover of lakes GT1 and GT2 (deep; group 1) and lake IWT1 (shallow; group 2) showed an inverse thermal stratification with bottom water temperature reaching 0.6°C (lake IWT1), 1.4°C (lake GT1) and 2.0°C (lake GT2; Fig. 6). DO was much lower in lake IWT1 (13% saturation below the ice cover, decreasing to ~1% near

sediment at 3.6 m depth in 2015) than in lake GT1 (43% below the ice, <1% below 6 m depth in 2015) and lake GT2 (no data in spring 2015; 70% below the ice, 26% at 9 m depth in 2016, but < 2% by mid-May in 2019; no data in spring 2015). Quickly after the ice cover melted at the beginning of July, the water column became weakly stratified in all three lakes, and warmer in lake IWT1 (above 8°C at the surface, as compared to ~5°C at the surface of the deeper lakes). By then, the water column already showed signs of DO depletion in the deeper lakes (GT1, GT2) but the summer stratification period was short, lasting

for about a month in the larger lake GT2. At the bottom of lake GT2, oxygen depletion occurred as soon as stratification established from the beginning of July and decreased down to 84% of saturation by mid-August 2015 (65% in 2018), until the autumnal turnover increased saturation level up again (unpubl. data). While only weak hypoxia was encountered at the bottom of lake GT2 in late summer, anoxia was reached in the hypolimnion of lake GT1 (Fig. 6). On its margin, lake IWT1 was generally well-mixed during the open-water period, but presented weakly stratified periods during warm and calm days (e.g.

on 3 August 2016). Early August profiles indicate that the entire water column was above 13°C in the shallow thermokarst lake IWT1, while the surface of lake GT1 and GT2 was slightly colder (respectively ~10°C and 11°C in 2016).

## 5. Discussion

The ice-marginal permafrost environment in the Qarlikturvik Valley is highly heterogeneous, as ground ice types and content can vary and coexist over short distances, leading to significant small-scale differences in lake types, in their morphological and limnological conditions, as well as their vulnerability to climate drivers and disturbances. The bathymetric data revealed the coexistence of two types of lakes with different morphological characteristics. We also found that different sedimentary facies were present in the cores collected from each group, suggesting different origins and evolutionary conditions.

### 5.1 Lake morphology and sediment stratigraphy of shallow thermokarst lakes formed in ice wedges

Sixty-two percent of the lakes in Qarlikturvik Valley are shallow (~2-4 m) and relatively flat at their bottom, with a central deeper pool. This group of shallow lakes displays maximum depths very similar to those in 'classic' thermokarst lakes (~1-4 m deep) that developed in segregation ice and ice wedge polygonal terrain, excluding lakes formed in Yedoma-type permafrost (Hinkel et al., 2012; Kanevskiy et al., 2014). Their depth is controlled by the depth of syngenetic ice wedges, and by the amount and distribution of ground ice in the substrate (Grosse et al., 2013). Their maximum depths are also in accordance with the thickness of the peaty silt sequence (~2-3 m) forming the surrounding material, which developed during the Late Holocene (Fortier et al., 2006). Subsequent thermokarst evolution in those basins is not likely to result in substantial subsidence of the lake or basin floor, which can be inferred from the moderate to low ice content of the lowermost glaciofluvial stratigraphic unit (A) of lake IWT1. The lake has been slowly expanding in the frozen silt-peat terrace, and thawing has reached the underlying glaciofluvial sand (Bouchard et al., 2020). The intermediate unit (B) includes a layer (~35-55 cm) of convoluted horizons, which is absent in lake GT1, and likely originates from collapsed bank material in response to thermo-mechanical erosion processes or disturbed horizon due to lake bottom subsidence after ground ice melting. The sediment profile from lake IWT1 is very similar to those found in lakes initiated by the degradation of ice-wedge and intrasedimental ice. These lakes typically have a transitional organic-rich layer containing peat derived from permafrost thawing and subsidence, underlying a layer of laminated organic-rich lacustrine mud (Biskaborn et al., 2013; Bouchard et al., 2017; Farquharson et al., 2016; Murton, 1996). Such an interpretation is further supported by the fossil diatom record investigated in lake IWT1 (named *Gull Lake*; Bouchard et al., 2020), showing a few species typical of cold, oligotrophic and organic-poor (e.g. glaciofluvial) streams in the bottom section (unit A), then dominated by diatom species typical of moss/peat substrates in the middle section (Unit B), while showing lacustrine conditions with more diverse habitats (benthic/tychoplanktonic taxa) in the upper section (Unit C).

### 5.2 Lake morphology and sediment stratigraphy of deep glacial thermokarst lakes formed in buried glacier ice

The other lakes (38%) stand out by their notably deeper basins (~ 5-12 m), and in some cases the presence of multiple sub-basins (e.g., lake GT2). Owing to the size of the lake depressions and their location adjacent to mounds of ice-contact deposits, these deeper lakes were primarily formed by the melting of buried glacier ice. This interpretation is supported by the presence

of two exposures of glacier ice revealed by lakeside slumps, which also indicates that the shoreline of lake GT2 is still ice-cored by glacier ice. Côté et al. (2010) have also reported the coexistence of shallow lakes and deeper lakes in similar depositional environments of the Qarlikturvik and the adjacent valley of glacier C-93, where nearly half the lakes had depths greater than 5 m (Fig. 2b; depth range: 5-21 m). The cryostratigraphic context of the Qarlikturvik valley is not conducive to the formation of deep depressions. The uppermost unit consists of interstratified peat and silt (thickness ~ 2–4 m) with high volumetric ice content (74.6 ± 10.6% ;Veillette et al., unpublished data), while the underlying unit is glaciofluvial sand and gravel, which typically have low excess-ice contents and minimal expected settlement. Given the thickness of the silt and peat sequence and low ice content of the glaciofluvial sands, the amount of intrasedimental ice, especially segregated ice in excess of the porosity, is not sufficient to create lakes with depths reaching up to 12 m, even if all the intrasedimental ice melted and the resulting water drained out of the soil porosity. In addition, the deepest sections of some of these lakes, for example lakes GT1 and GT2, are ~ 5 m below current sea level, indicating burial in a glaciomarine/glaciofluvial environment followed by isostatic uplift.

The glacial origin of these deep lakes is further corroborated by the analysis of the sediment core collected in lake GT1 (12.2 m), which differs significantly from the one obtained in lake IWT1 (Fig. 5). In lake GT1, four depositional stages were inferred from the sediment profiles. The inception of lake GT1 began with the collapse of supraglacial material during melt-out of stagnant glacier ice, which resulted in re-sedimentation of sand and gravel from glaciofluvial and mass movement deposits into a forming basin. Inclusions of fibrous peat in unit A and the ~10 cm darker layer (Unit B) of organic debris and inorganic material (mostly silt and sand) were probably derived from surficial vegetation on upland surfaces washed in as the lake basin developed. This interpretation is further confirmed by the presence of scattered, although identifiable, diatom species typical of this depositional environment (e.g., *Eunotia*, *Cymbella*, *Pinnularia*). This basin was then filled by sandy mud (Unit C) deposited by the combined action of meltwater streams and aeolian activity. The most prominent features of this core from lake GT1 are the sharp boundaries of units B and C, indicating a shift in the depositional conditions, which do not reflect gradual deposition within a stable lake floor (Henriksen et al. 2003). Like the upper core section in lake IWT1, the upper unit D includes recent lacustrine sediments composed of organic-rich mud (gyttja). This material was deposited within deeper, calmer waters where fine material can settle. The laminations reflect variability in minerogenic inputs and are likely related to terrestrial runoff or aeolian activity. Similar stages of sedimentation were identified from glacial thermokarst lake basins, including one lake with multiple sub-basins (up to > 14 m deep) in the continuous zone of permafrost in northern European Russia, where buried glacier ice has survived for ca. 80 000 years from the last glaciation (Henriksen et al., 2003), or in older glacial lakes in northern USA (Yansa et al., 2020). Reversal of ages in the core suggests that older organic matter was also washed into the lake during the mid-Holocene, causing abnormally old dates in basal core sediments, a common dating problem in high-latitude lakes (Bouchard et al., 2017; Wolfe et al., 2004). Furthermore, the absence of identifiable diatom taxa (i.e. presence of scarce fragments only) in the upper section of the core (specifically Unit A) is puzzling. The relatively high pH (>

8) in this lake, especially during the spring bloom (pH ~ 10), could help explain the poor preservation of diatom valves (Ryves et al., 2006).

Moreover, our spatial analysis demonstrated that lake distribution is strongly linked to the maximum and recessional positions of local mountain glaciers and the LIS in both Qarlikturvik Valley and the southern plain of Bylot Island. This is particularly evident in the valley, as shown by the three well-defined lake clusters (Fig. 2a). For most of the southern lowland coastal plains, moderate to high point densities were also encountered within the extent of the LIS. There is also a notable increase in lake density close to the contemporary ice margin, which we interpret to be the result of a relatively recent and continuous deglaciation process. We found that, even after accounting for landscape heterogeneity (i.e. high slope gradients, bedrock exposures), the lakes are still far more clustered when compared to a random spatial distribution. As a result, we propose that the clustering reveal patterns caused by the presence of patches of buried glacier ice. This provides additional evidence for supporting the glacial origin of these lakes. The presence of deep lakes and numerous thaw slumps in Qarlikturvik Valley indicates the delayed melting of several bodies of buried glacier ice as compared to the Holocene glacier retreat. The ice-free zones of Bylot Island are therefore still strongly influenced by its glacial legacy given the presence of late Pleistocene-age glacier ice buried in the permafrost in Qarlikturvik Valley (Coulombe et al., 2019) and in other valleys and coastal plains of the island (Klassen, 1993; Moorman and Michel, 2000).These ice-cored landforms have been adjusting to non-glacial conditions and their evolution is strongly linked with geomorphological processes and local terrain conditions and stability.

**5.3 Conceptual model of thermokarst lake development in ice-wedges polygon terrain and buried glacier ice**

Several studies have described the stages of thermokarst lake development and thaw lake cycle in permafrost environments, such as in the Yedoma (Morgenstern et al., 2011; Shur et al., 2012), lacustrine environments with ice wedges aggradation and degradation in the basins (Billings and Peterson, 1980; Jorgenson and Shur, 2007), and ice-rich cryogenic mounds (Calmels et al., 2008), or ice-wedge and intrasedimental ice (Czudek and Demek, 1970; Jorgenson and Osterkamp, 2005). In a previous study, Bouchard et al. (2020) presented a four-stage conceptual model for lake IWT1 (named *Gull Lake*) that describes thermokarst inception and evolution in syngenetic ice-wedge polygon terrain during the Holocene. Based on this model, lake IWT1 developed in a pre-existing topographic depression (~1-2 m) that collected snow and meltwater (stage 0, initial conditions). The first phase of thermokarst started at around 2100 BP in response to active layer deepening and ice wedge melting, which initiated the development of small and shallow ponds over the degrading ice wedges (stage 1). Thermokarst ponds started to coalesce with neighbouring water bodies over and at the edge of ice-wedge polygons to form a small lake (stage 2). Over time, this lake expanded in the ice-rich polygon terrace because of surface permafrost degradation via lateral thermal erosion and vertical thaw settlement and consolidation in the ice-rich silt-peat terrace, and eventually in the underlying glaciofluvial sediments (stage 3). The last stage suggests a possible long-term future scenario where the lake disappears through the gradual gyttja accumulation and lake infilling or lake drainage, which can sometimes be catastrophic (Bouchard et al.,

2020 and citations therein). The conversion of these aquatic ecosystems to terrestrial or wetland ecosystems is usually followed by a reactivation of old ice wedge networks or growth of pingos as permafrost aggrades in unfrozen drained lake deposits once exposed to cold temperatures, which eventually begin a new phase of the thaw-lake cycle (Billings and Peterson, 1980; Mackay and Burn, 2002; Jorgenson and Shur, 2007).

Based on the geomorpholology of the deeper lakes and lake sediment profiles of lake GT1, we also developed a four-stage model of glacial thermokarst lake formation and evolution in the specific stratigraphic context of buried glacier ice within the study area (Fig. 7).

**Stage 0: Initial conditions.** During the Last Glacial Maximum, the LIS and local ice caps covered much of the Qarlikturvik Valley, and many outlet glaciers were channelled through major valleys of Bylot Island and terminated on the lowlands.

**Stage 1: Burial of glacier ice.** Beyond the active margins of local glaciers or the LIS, wide areas of glacier ice were likely buried *in situ* by glacigenic sediments transported and reworked on top of an active or stagnant glacier margin by mass movement and meltwater (Fig.7a). The burial of glacial ice can still be seen today at the margins of many glaciers on Bylot Island. The TC brightness index exhibits a strong positive trend at the glacier margins, indicating dryer and unvegetated surfaces. This corresponds to sediment accumulation onto the glacier surface and represents a modern analogue of the burial of glacier ice. The brightness trend correlates well with the active burial of ice observed at numerous locations at the margins of glaciers C-93 and C-79 (Fig. S4). Progressively, stagnant ice blocks became isolated from the upper active flowing ice. On Bylot Island, bodies of glacier ice were preserved at various places in the outwash plain, in mounds of ice-contact stratified drift, and moraines (Coulombe et al., 2019; Klassen, 1993). Interpretations of the sedimentary sequence overlying the buried ice studied in the valley indicated that the burial of the ice involved glaciofluvial deposition directly on the ice, which was followed by plant colonization. This situation can occur during or after postglacial isostatic uplift. In some cases, glaciofluvial sands and gravels were also covered by colluvial sediments as debris were transferred away from topographic highs by mass movements and meltwater (Coulombe et al., 2019). Preservation of the ice for several millennia was possible because the sediment cover became sufficiently stable and reached or exceeded the active layer thickness, but also because the neoglacial climatic conditions during the second half of the Holocene were conducive to syngenetic permafrost aggradation following glacier retreat (Fortier et al., 2006; Fortier and Allard, 2004; Bouchard et al., 2020).

**Stage 2: Initiation of buried glacier ice melting.** Melting of the upper glacier ice and formation of a depression begins when a deepening active layer reaches the glacier ice (Fig. 7b). Local factors such as topography, thickness of the active layer, snow accumulation and water pooling in pre-existing depressions, as well as thermal properties of soil all play a role in initiating ice melting. We suggest three scenarios of initial pond inception following the burial of glacier ice: (1) ponds may have formed quickly in the proglacial environment or later during deglaciation as water pooled in these pre-existing topographic depressions caused by the uneven ablation of the glacier surface (i.e. differential melting of dead ice); (2) the ice was first buried under a

thin cover of glacigenic sediment, which was close to the active layer thickness. The sediment cover was thick enough to slow down the melting rate of underlying ice without completely preserving it, which allowed a gradual melting of the ice creating small depressions; (3) the ice was buried under thick sediment cover, acting as a barrier to heat transfer, and preserving the ice in the long term. However, unusual warm and wet conditions have periodically caused the active layer to deepen considerably, initiating the melting of the underlying ice and creating new depressions. Following the burial of the ice, the uneven ablation of the glacier surface produced an irregular topography of ridges and mounds. Since buried glacier ice is still present in the study area, thaw slump activity is thought to have been a fundamental driver of its degradation by exposing the ice and accelerating its melting (Coulombe et al., 2019).

**Stage 3: Lake inception and syngenetic ice growth**. Episodes of warming sufficient to cause degradation of the existing permafrost and the buried glacier ice in the valley triggered thermokarst lake initiation (Fig. 7c). Summer-melt layers from the Agassiz Ice Cap and Greenland ice sheet provide robust records of warmer events during the Holocene (~7-8 ka; Fisher et al., 1995; Westhoff et al., 2021). These warmer periods likely initiated or accelerated the ice melt when (1) a thin layer of sediments covered the ice, or (2) topographic depressions allowed the accumulation of snow and water, hence overall warmer conditions, which further accelerated the melting of the buried glacier ice. This resulted in subsidence of the terrain surface, deeper snow accumulation in winter and ponding of surface water during the warm season, which began to thaw the underlying permafrost. During the first phase of lake development, relict glacier ice can serve as a focal point for the onset of accelerated thermokarst degradation. If exposed, the ice core then undergoes accelerated wastage through the effects of solar radiation or becomes buried again under the slumping material until a new thermal balance can be reached. We cannot deduce absolute timing for the inception of lake GT1 since no reliable basal dates are available. However, we suggest that lake inception of these deeper glacial lakes occurred sometime during the mid-Holocene and preceded that of the shallowest lakes formed by the thawing of ice wedges, which were [14]C dated at around 2100 yr BP (Bouchard et al., 2020).

**Stage 4: Thermokarst lake inception and lake expansion within permafrost.** Once a lake gets deeper than the maximum thickness of the winter ice cover (~2 m ± 20 cm in the valley as measured by Prėskienis et al., 2021), it will continue to grow laterally (thermo-mechanical erosion) and vertically (subsidence) by thermokarst processes each year (Fig.7d). The water sensitive index TC-wetness exhibits a moderate to strong positive trend for many lakes in the valley, driven by the gradual erosion lake shores containing ice-rich permafrost. The rate of expansion depends on the local climatic conditions, ground-ice content and lake bed temperature. In cases where buried glacier ice remains present beneath the lakebed, the ice will slowly continue to melt, causing lake bottom subsidence. Further ground ice melting and the resulting thaw slumps contribute to lake expansion, as shown by the head scarps located close to the shoreline of lake GT2 (Fig. 4). Other studies have shown that thaw slumping is an important mechanism of lake expansion (Hinkel et al., 2012; Plug and West, 2009; Kokelj et al., 2009). Our results indicate that these glacial thermokarst lakes also evolved at a later stage as 'classic' thermokarst lakes that are now slowly expanding in area and volume, because of the melting of intrasedimental ground ice and ice wedges in the frozen silt-

peat terrace and in the underlying glaciofluvial and till material. The shorelines of glacial lakes are expected to be very smooth and roughly circular or oval-shaped, as shown by the morphological analysis of glacial lakes located at the termini of glaciers C-93, C-79 and C-67. However, most lakes studied here display slightly irregular shorelines (Fig. S5). In the Qarlikturvik Valley, the shoreline morphology of the deeper glacial thermokarst lakes is very similar to the other thermokarst lakes, indicating that all lakes are now laterally expanding in the polygon terrace by thermal and mechanical erosion. Thermokarst is an active landscape change mechanism currently operating in the valley and on the island in general (Bouchard et al., 2020; Fortier et al., 2007; Godin et al., 2014). Today, the lakes can expand by thermal subsidence and different shoreline erosional processes including: (1) the development of thermo-erosional niches; 2) the mechanical erosion caused by lake ice pushing against the shore, and 3) the incorporation of adjacent polygonal ponds into the lake (Jones et al., 2011). Eventually, the lakes may cease expanding in the event of partial surface/subsurface drainage through various permafrost degradation processes. Complete drainage of these glacial thermokarst lakes remains is impossible due to their great depth which is below the base level of streams and river and even below sea level in some instances. This situation allows these lakes to persist over time, unlike shallow lakes that have developed in segregation ice and ice-wedge polygons, which are susceptible to complete drainage (Mackay, 1992) and to return to terrestrial conditions. This shows the interplay of climatic (external) and local landscape (internal) processes in the inception and evolution of thermokarst lakes in general, including the ones developed through melting of buried glacier ice.

## 5.4 Implications on Arctic lakes ecosystem dynamics

Lake morphometry, specifically depth, plays an important role in regulating lake water temperature and associated biogeochemistry. It influences the mixing regime and the number of thermal overturn events per year during the open-water period (i.e., if the lake is monomictic, dimictic or polymictic, the latter being more common for Arctic lakes; Rautio et al., 2011). This is intrinsically linked to water column aeration and light regime, thus exerting a strong control on respiration and primary production (Vincent, 2010). Results indicate that the three studied lakes can be considered as cold polymictic (or potentially dimictic depending on the year for lake GT1, although a mooring would be needed to validate this). Among the three lakes studied for their limnological characteristics, bottom water of shallow lake IWT1 was the coldest by late winter but the warmest by late summer, a pattern directly linked to its mixing regime where meteorological conditions are more likely to influence bottom water temperature and talik formation. We also found that lake morphology influences dissolved oxygen. Lake IWT1 showed the lowest oxygen concentration at the end of the winter (< 2 mg L$^{-1}$ or < 13% saturation in 2015), likely linked to its large sediment area to water volume ratio and its higher organic content at the lake bottom (submerged peat polygons, as opposed to less organic sediments in the deeper lakes GT1 and GT2) leading to a faster depletion of oxygen (Vincent, 2010; Ward et al., 2017). In addition of controlling GHG cycling, this can be a significant limiting factor for overwintering fish populations (Leppi et al., 2016). For the deeper lakes (GT1, GT2), the difference in water column stability controlled bottom oxygen saturation levels during the open-water period, which decreased well below 60% in lake

GT1 (reaching anoxia just above sediment), while it always remained above this level in lake IWT1. The stronger gradient in lake GT1 is likely related to its smaller size (smaller fetch) and greater depth. Climate change may therefore not only affect water temperature, mixing regime and oxygen availability through warming and summer lengthening, but also through effects on the evolution of lake morphology from the melting of buried glacier ice.

These differences in the mixing regime and oxygen availability, controlled by lake morphology (size and especially depth), exert a strong control on the timing (seasonal differences) and magnitude of GHG emissions ($CH_4$, $CO_2$, and their relative proportion) from the water column to the atmosphere (Prėskienis et al., 2021; Bouchard et al., 2015a; Hughes-Allen et al., 2021; Matveev et al., 2016). Previous studies showed that lake IWT1 generally maintained high GHG fluxes during the open-water period as compared to the deeper lakes GT1 and GT2 (named *kettle lakes* Prėskienis et al., 2021). Once more, the combination of warmer temperatures and higher organic content of lake IWT1 likely explains its higher GHG emissions (BYL66, Prėskienis et al. 2021; see Figs. 3 and 4, and Table 4), averaging 27.1 mmol $CO_2$ equivalent $m^{-2}$ $d^{-1}$, as compared to 10.8 mmol $CO_2$ equivalent $m^{-2}$ $d^{-1}$ from lake GT1 (mainly caused by differences in $CH_4$ ebullition; not assessed in lake GT2). Considering that most GHG are emitted from lake sediment (Bastviken et al., 2004), it is important to underline that the largest sediment area of a lake is in contact with epilimnetic (shallow) waters, and therefore not only bottom water temperature in the deepest pelagic section of a lake needs to be assessed. Moreover, glacial thermokarst lakes subjected to partial drainage will maintain year-round GHG emissions whereas thermokarst lakes formed in polygonal terrain and subjected to complete drainage will have a totally different GHG emissions regime (terrestrial GHG emission during the warm season when active layer soils are thawing). Due to the importance and diversity of lakes across the circumpolar Arctic, a better knowledge of their bathymetry and landscape variability is necessary to upscale local biogeochemical assessments to regional or continental scales. The future melting of buried ice, widespread in certain regions of the Arctic but overlooked, will form new lakes that will present different features than classic thermokarst lakes, notably in terms of water temperature, mixing regime, oxygen availability, GHG production, and GHG ages.

## 6. Conclusion

Spatial variability in ground ice conditions is an important factor driving lake inception, evolution and distribution on Bylot Island. This study confirms that glaciated permafrost terrain containing various types of ground ice, including buried glacier ice, can influence the spatial distribution of lakes, lake bathymetry, limnological properties as well as lake bottom morphology and sediment stratigraphy. The origin and growth of numerous thermokarst lakes in the Qarlikturvik Valley, Bylot Island, has been examined using bathymetric and field surveys, high-resolution remotely sensed imagery, and lake sediment analysis. Slightly more than half of the twenty-one studied lakes tend to be shallow (~2-3 m), while the other lakes stand out by their notably larger depths (~9-12 m). The stratigraphic analysis of two lake sediment cores revealed two distinct basin types in terms of sediment accumulation, although more work is needed to confirm this difference by collecting sediment cores from a

larger set of lakes in the Qarlikturvik Valley, and also in other glacierised Arctic tundra settings. These dissimilarities indicate that these lakes have a different origin and evolution throughout the Holocene as well as distinct depositional history and sedimentological signature. These results suggest that the melting of ice wedges and intrasedimental initiated the formation of the shallow lakes (< 5 m), while the melting of buried glacier ice has triggered the inception of the deeper lakes (> 5 m, up to 12 m) in the study area. The glacial origin of deeper thermokarst lakes is supported by the past and current presence of buried glacier ice as well as numerous stable and active thaw slumps in the study area,. In addition, the shallow and deeper lakes coexist within the same depositional environment, indicating that these lakes have been subjected to the same environmental and climatic conditions, and therefore notable depth difference must be related to different ground ice volume or time spanned since inception. Moreover, analysis of lake morphometry and distribution revealed that lakes are more densely distributed near the most recent ice positions. This suggests a relationship between the formation of lakes and the deglaciation patterns in both Qarlikturvik Valley and the broader southern plain of Bylot Island. Given future climate projections, it is likely that Arctic lowlands with glacier ice buried in permafrost will change dynamically because of surface permafrost degradation and melting of relict glacial ice. It is expected that the deepening of the active layer and talik development, as well as the enlargement of Arctic lakes in response to global warming, will reach undisturbed buried glacier ice. This will create new aquatic ecosystems and strongly modify existing ones through the lateral expansion of lakes caused by wind- and circulation-driven erosion, thaw slumping and thaw subsidence along lake margins. In turn, this will likely have pervasive effects on geomorphological, hydrological, and ecological processes of affected landscapes, including the high-latitude and global carbon budgets and oxythermal quality of fish habitats.

*Data availability.* The following related datasets are available in the Nordicana D collection at Centre d'études nordiques (CEN – Centre for Northern Studies) (http://www.cen.ulaval.ca/ nordicanad/). The complete citations of each dataset appear in the reference list of this article.

Fortier et al. (2022): "Morphometry of glacial lakes formed in front of glaciers C-93 and C-79, Bylot Island, Nunavut", https://doi.org/ 10.5885/45765CE-0DBCF1FE81114010.

Fortier et al. (2021): "Organic matter content and grain size distribution in a lake sediment core, Bylot Island, Nunavut, Canada", https://doi.org/10.5885/45603CE-21852993EE434926.

Fortier et al. (2021): "Radiocarbon (14C) dates in terrestrial and aquatic environments, Bylot Island, Nunavut", https://doi.org/10.5885/45651CE-C6FD628F45E44578

*Author contributions.* The study was conceived by SCo, DF and FB. SCo prepared the article with contributions from all co-authors. IL provided temperature and dissolved oxygen data and contributed to the analysis and interpretation of the data. SCh collected bathymetric data. RP and MP carried out the analysis and interpretation of diatoms and GPR, respectively.

*Competing interests.* The authors declare that they have no conflict of interest.

*Acknowledgements.* This project has been funded by the Natural Sciences and Engineering Research Council of Canada (NSERC), Fonds de Recherche du Québec – Nature et technologies (FRQNT), and the W. Garfield Weston Foundation. Additional support has been provided by ArcticNet, the Polar Continental Shelf Program (PCSP), the Northern Scientific Training Program (NSTP), and the NSERC Discovery Frontiers grant "Arctic Development and Adaptation to Permafrost in Transition" (ADAPT). The fieldwork benefited from the logistical support provided by Gilles Gauthier and his team (U. Laval). We gratefully acknowledge the hospitality and assistance of the community of Mittimatalik (Pond Inlet) and the staff of the Sirmilik National Park. Special thanks to Audrey Veillette, Vilmantas Prėskienis, Karine Rioux and Zhaoyi Zhang for their help in the field and laboratory, and to Denis Sarrazin for providing photographs and taking care of the moorings. We thank the editor, Regula Frauenfelder, as well as Steve Kokelj and a second anonymous reviewer for constructive comments that greatly helped to improve the final manuscript.

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

## Figures and tables

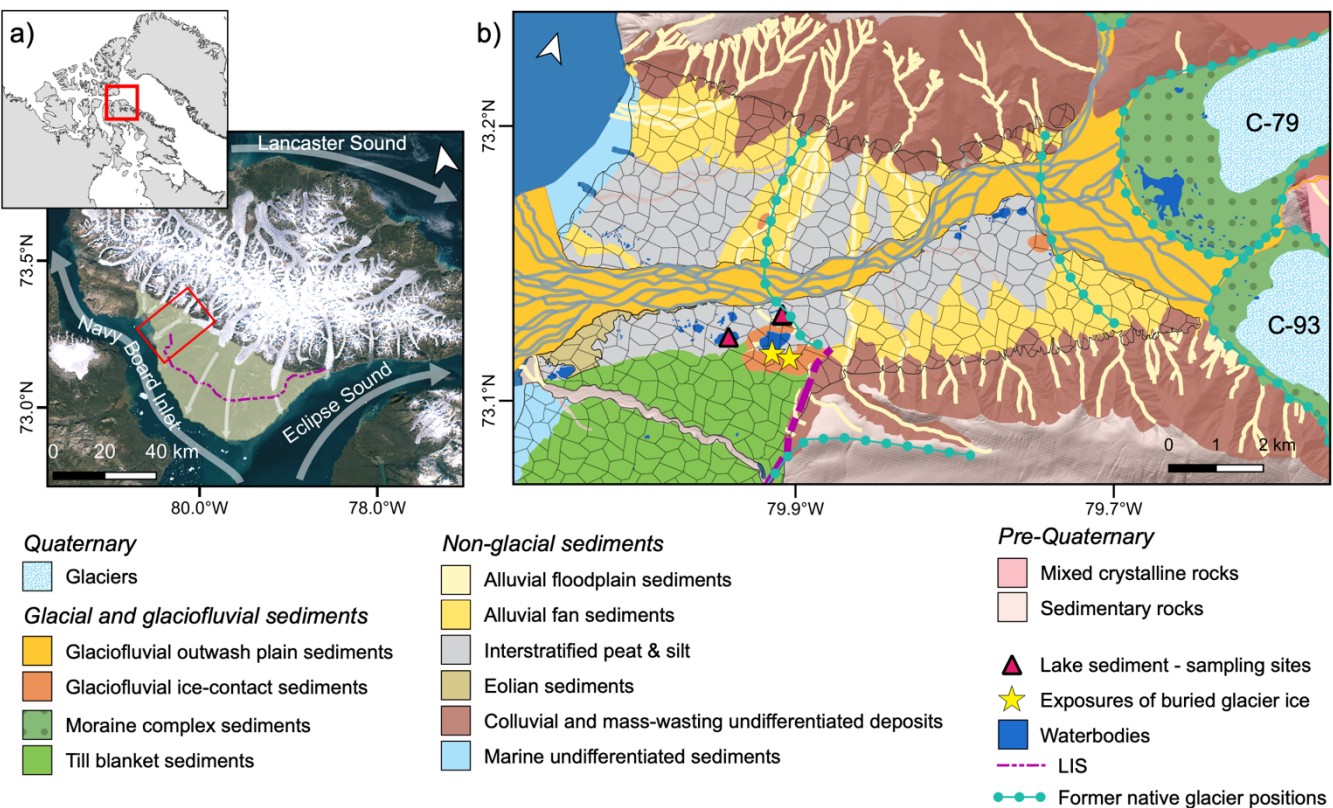

**Figure 1: a) Location of Bylot Island, Nunavut, Canada and the study area in the Qarlikturvik valley (background: NRCan Landsat-7 orthorectified mosaic, 3 August 2010). The shaded area shows the southwestern plain of Bylot Island, b) Surficial geology of the valley and location of the sampling sites. The net pattern represents the polygonal patterned ground. The white arrows show the direction of ice flow within and around Bylot (Margold et al., 2015).**

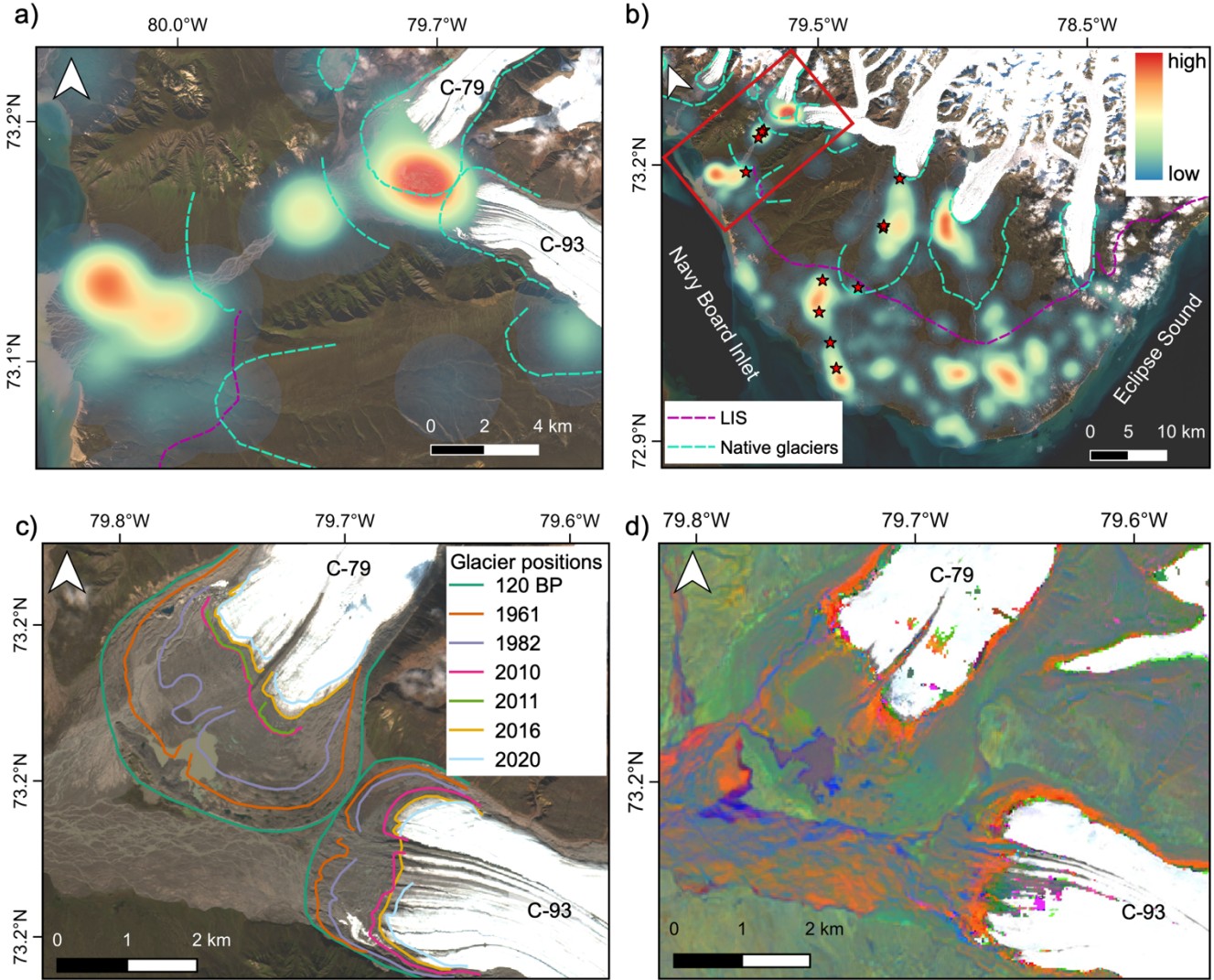

**Figure 2: a) Spatial point density of lake locations in the Qarlikturvik Valley, b) Spatial point density of lakes in the southern plain of Bylot Island. The blue dashed line shows former limits of local mountain glaciations. The purple dashed line shows the limit of the Laurentide Ice Sheet (LIS) as defined by Klassen (1993). The red stars indicate the locations of deep lakes (between 5 m and 21 m) studied in Coté et Pienitz (2010). (Background: Sentinel-2 (ESA) image courtesy of the Copernicus Open Access Hub), c) Glacier terminus positions from 120 yr BP to present and d) Tasselled Cap transformation images obtained for Google Earth Engine (Gorelick et al., 2017) The accumulation and movement of sediments in the outwash plain and at the glacier front are represented by red and orange colours on the images (dry and unvegetated areas; TC brightness). Wetter areas, such as eroding cliff, lake shore, or river channel are shown in blue (high TC wetness). Vegetated areas are distinguished by teal and yellow colours (TC greenness).**

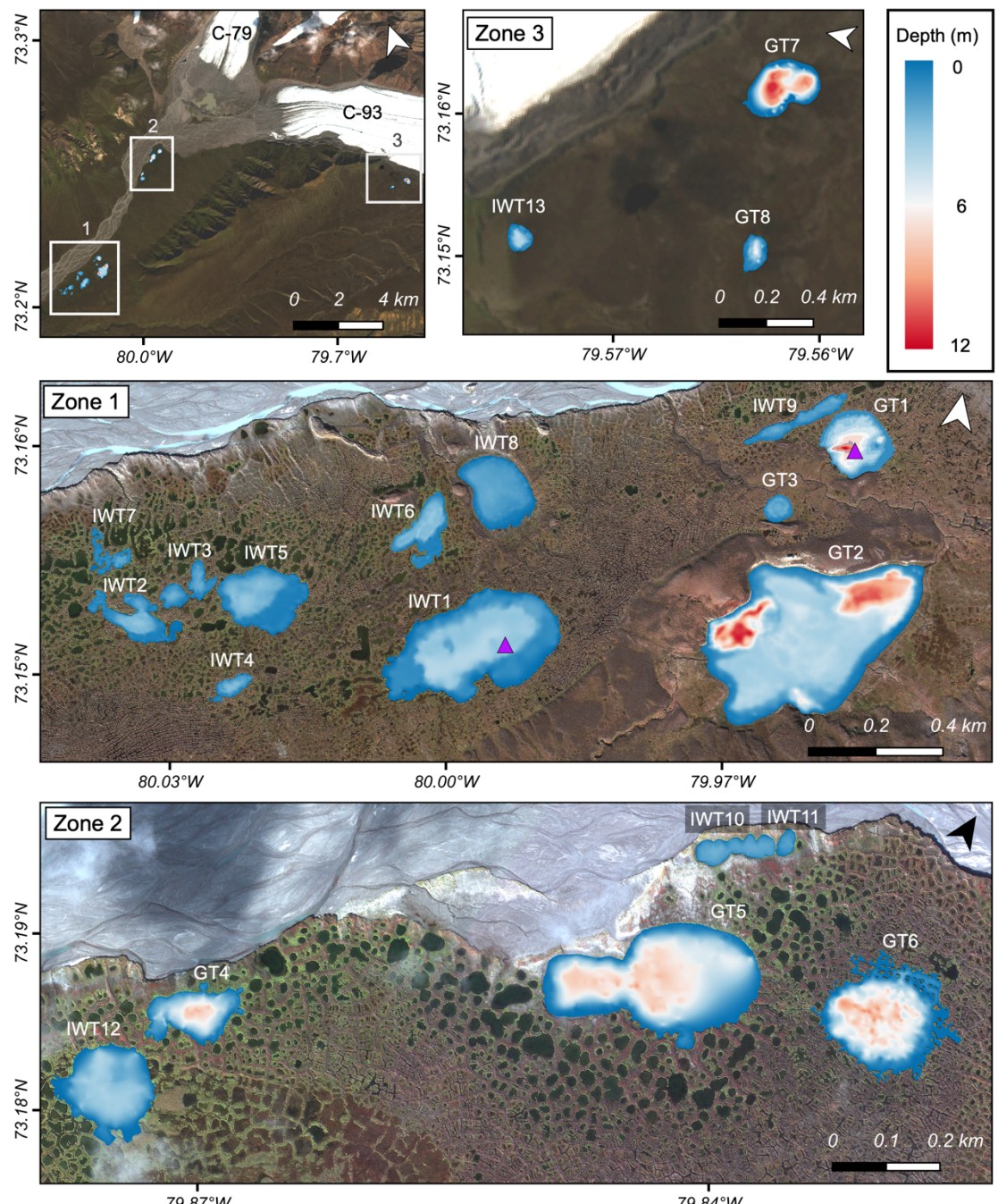

**Figure 3: Bathymetric maps of the 21 lakes surveyed in the Qarlikturvik valley (background: GeoEye, 2010). The yellow stars show the location of massive ice exposures. Purple triangles on lakes IWT1 and GT1 indicate sediment coring locations. Glacial thermokarst lakes (max. depth > 4 m): GT 1 to 8; Ice-wedge thermokarst lakes (max. depth < 4 m): IWT1 to 13**

**Table 1. Characteristics of lakes (n=21) for which bathymetric data were collected in 2015.**

| Zone | Name | Latitude | Longitude | Surface Elevation (m a.s.l.) | Max. depth (m) | Depth - Basin (m) Mean | Std. dev. | Depth - Platform (m) Mean | Std. dev. | Area (m²) | Perimeter (m) | Shoreline development index | Other names | Elongation ratio | Distance to glacier margin (m) |
|---|---|---|---|---|---|---|---|---|---|---|---|---|---|---|---|
| | IWT1 | 73.153 | -79.999 | 8.4 | 4.1 | 2.9 | 0.8 | 1.3 | 0.4 | 10076.6 | 694.9 | 2 | BYL66 | 1.2 | 464 |
| | IWT2 | 73.152 | -80.030 | 7.1 | 3.2 | 2.0 | 0.6 | 1.1 | 0.3 | 19367.8 | 1111.2 | 2.3 | | 1.5 | 1444 |
| | IWT3 | 73.153 | -80.026 | 7.0 | 2.4 | 1.5 | 0.4 | 0.9 | 0.2 | 30968.2 | 709.7 | 1.1 | | 0.4 | 1280 |
| | IWT4 | 73.151 | -80.021 | 8.8 | 2.8 | 1.9 | 0.5 | 1.2 | 0.3 | 115306.2 | 1689 | 1.4 | | 1.2 | 1197 |
| | IWT5 | 73.153 | -80.019 | 7.0 | 3.3 | 2.2 | 0.6 | 0.9 | 0.2 | 37549.1 | 926.7 | 1.3 | | 0.6 | 1064 |
| 1 | IWT6 | 73.156 | -80.006 | 7.7 | 4.3 | 2.3 | 0.6 | 1.7 | 0.2 | 18379.7 | 873.3 | 1.8 | | 1 | 560 |
| | IWT7 | 73.153 | -80.034 | 7.1 | 2.9 | 1.5 | 0.4 | 1.2 | 0.3 | 6470.4 | 952.6 | 3.3 | | 0.8 | 1505 |
| | IWT8 | 73.157 | -80.000 | 5.7 | 2.5 | 1.5 | 0.3 | 0.8 | 0.2 | 34167 | 790.5 | 1.2 | | 0.8 | 329 |
| | IWT9 | 73.160 | -79.973 | 8.3 | 3 | 1.5 | 0.3 | 1.1 | 0.2 | 11501.6 | 790.1 | 2.1 | BYL123 | 3.8 | 586 |
| | GT1 | 73.160 | -79.968 | 7.9 | 12.2 | 5.4 | 1.5 | 3.6 | 1.0 | 5226.1 | 447.4 | 1.7 | BYL36 | 0.7 | 491 |
| | GT2 | 73.155 | -79.969 | 8.2 | 11.7 | 4.7 | 2.0 | 2.2 | 0.6 | 209426.6 | 1991.9 | 1.2 | BYL37 | 1.1 | 514 |
| | GT3 | 73.158 | -79.974 | 6.8 | 6.1 | 1.8 | 0.3 | 1.0 | 0.2 | 5264.5 | 272.9 | 1.1 | | 0.7 | 491 |
| | GT4 | 73.185 | -79.870 | 15.1 | 8.4 | 4.2 | 1.8 | 1.6 | 0.4 | 12963.4 | 695.3 | 1.7 | | 1.1 | 1726 |
| | GT5 | 73.190 | -79.848 | 15.1 | 9.4 | 4.7 | 2.1 | 0.8 | 0.2 | 59699.7 | 1190 | 1.4 | | 1.1 | 868 |
| 2 | GT6 | 73.191 | -79.836 | 18.3 | 9.8 | 5.0 | 2.1 | 2.4 | 0.3 | 35566.3 | 2673.8 | 4 | | 0.8 | 441 |
| | IWT10 | 73.192 | -79.849 | 16.9 | 2.3 | 1.5 | 0.5 | 0.5 | 0.1 | 5354.2 | 403.7 | 1.6 | | 1.7 | 768 |
| | IWT11 | 73.193 | -79.847 | 17.0 | 1.8 | 1.4 | 0.3 | 0.8 | 0.2 | 1292.5 | 139.8 | 1.1 | | 1 | 674 |
| | IWT12 | 73.183 | -79.872 | 14.2 | 3.9 | 2.6 | 0.8 | 0.9 | 0.2 | 22894.5 | 817.8 | 1.5 | | 0.7 | 1873 |
| | GT7 | 73.144 | -79.531 | 354.0 | 15.4 | 6.9 | 4.0 | 1.0 | 0.3 | 54464.4 | 944.1 | 1.1 | | 1 | 256 |
| 3 | GT8 | 73.144 | -79.554 | 346.5 | 5.9 | 2.6 | 1.4 | 0.9 | 0.3 | 12754.9 | 446.4 | 1.1 | | 0.6 | 829 |
| | IWT13 | 73.153 | -79.558 | 359.4 | 4.3 | 2.6 | 1.0 | 1.2 | 0.3 | 9757.7 | 367.4 | 1 | | 0.8 | 380 |

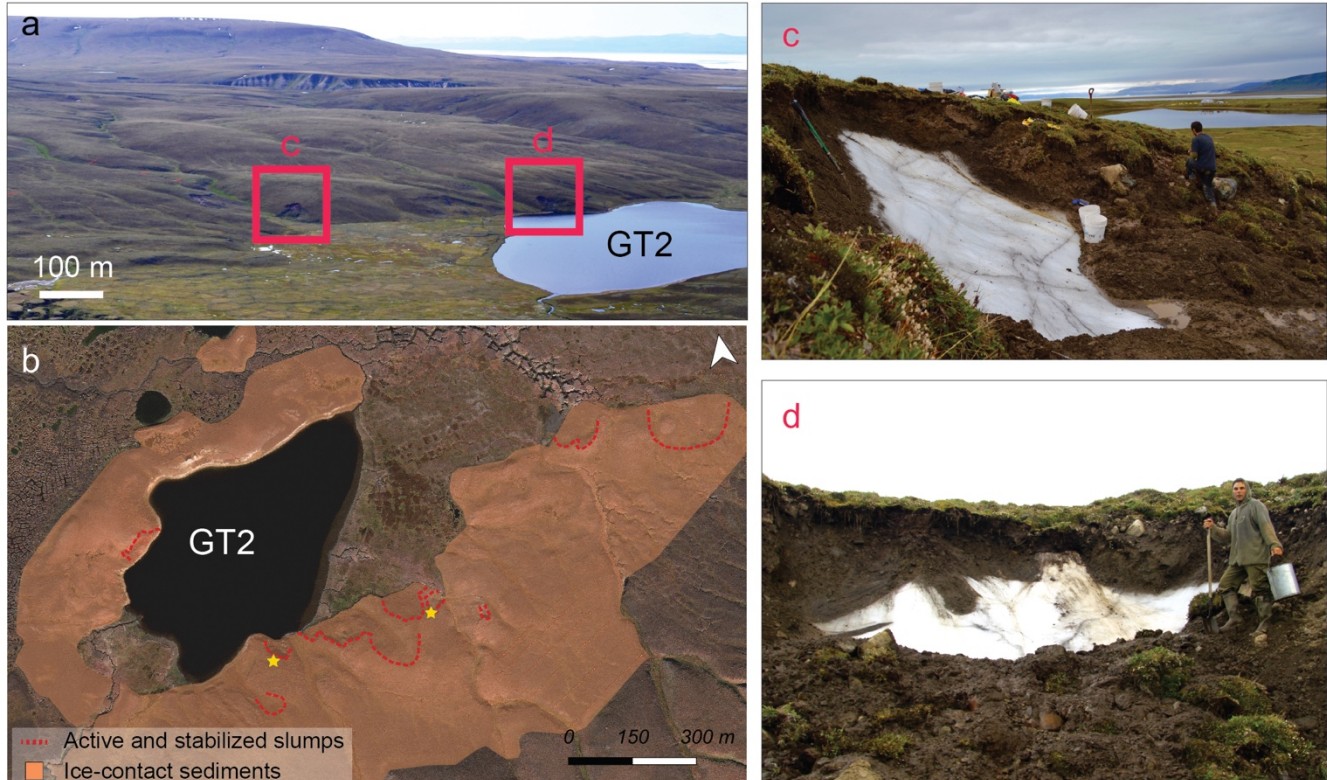

**Figure 4: a)** Aerial view of two exposures of buried glacier ice located nearby glacial thermokarst lake GT2, **b)** Map showing the distribution of active and stable thaw slumps nearby glacial thermokarst lake GT2 (background: GeoEye, 2010), **c)** We interpreted the massive ground-ice, exposed at the headwall of thaw slump, as buried glacier ice on the basis of cryostratigraphic, crystallographic and geochemical analyses (Coulombe et al., 2019), **d)** This massive ice exposure was not studied in detail since the ice had been buried again under a thick cover of slump material. However, the ice displays a very similar appearance to the first exposure located one hundred meters away (Photo courtesy of Denis Sarrazin).

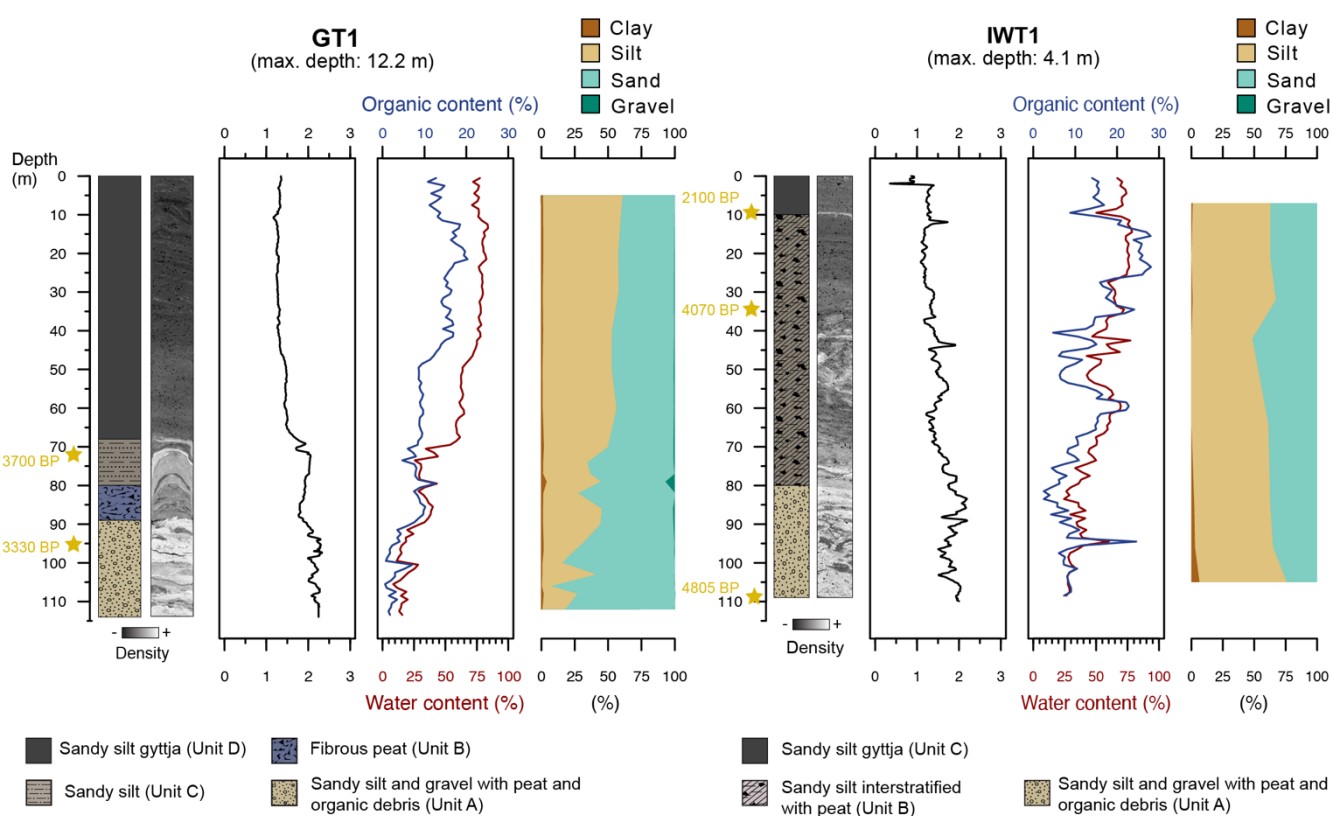

**Figure 5: General stratigraphy of cores sampled in lake GT1 (deep glacial thermokarst lake, max. depth = 12.2 m) and IWT1 (shallow ice wedge thermokarst lake, max. depth: 4.1 m).**

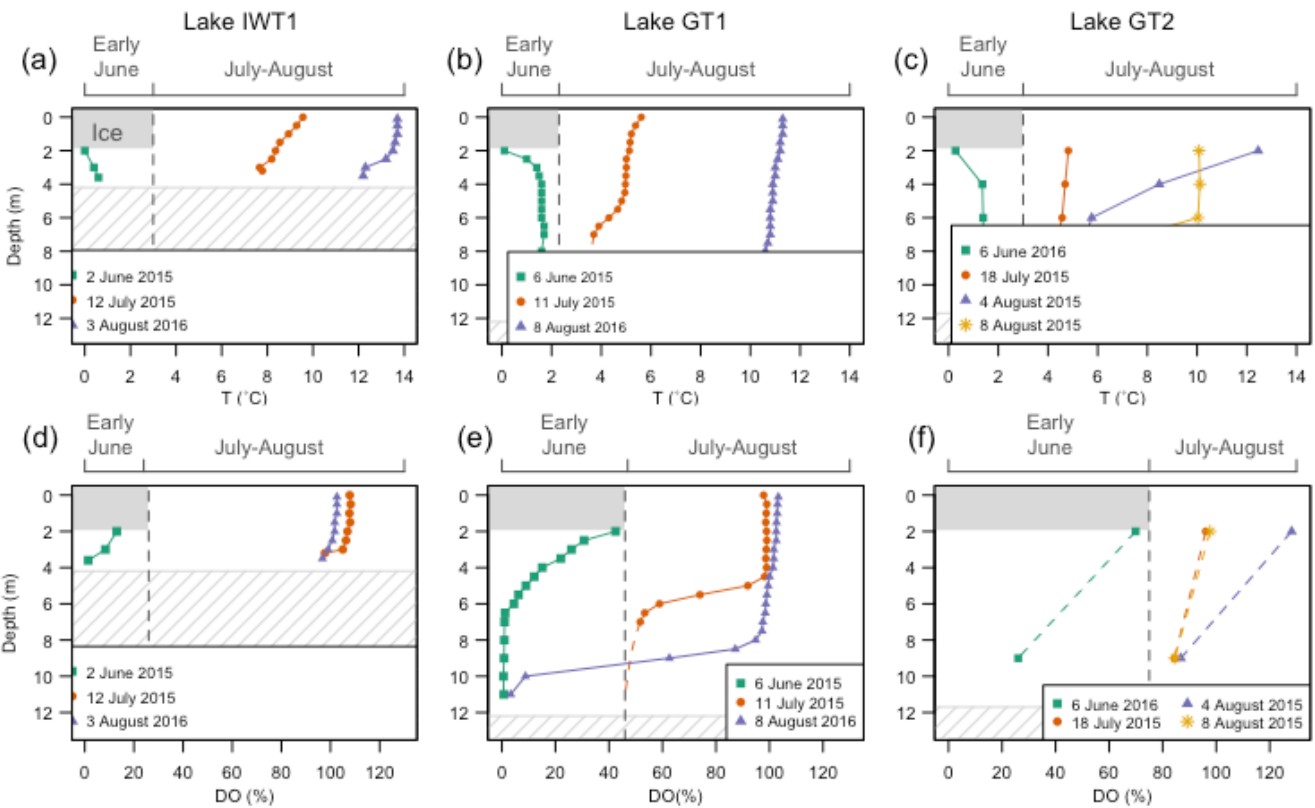

**Figure 6: Early June, July and August temperature and dissolved oxygen (DO) profiles for ice wedge thermokarst lake IWT1 (a and d), and glacial thermokarst lakes GT1 (b and e) and GT2 (c and f). The grey rectangle represents the ice cover in early June) and the average maximum ice thickness is 2 m ± 20 cm (measured in 2015 and 2016; Preskienis et al., 2021). For lake GT1 (b and e), the dashed lines are simply connecting the two available data (from the mooring) but the shape is likely to follow the more detailed temperature profiles. The hatched rectangle indicates the lake bottom. Note that the dates are slightly different in lake GT2. Profiles from lakes IWT1 and GT1 in early June and late summer (August) are adapted from Preskienis et al. (2021; respectively corresponding to lakes BYL66 and BYL36), allowing to compare with lake GT2.**

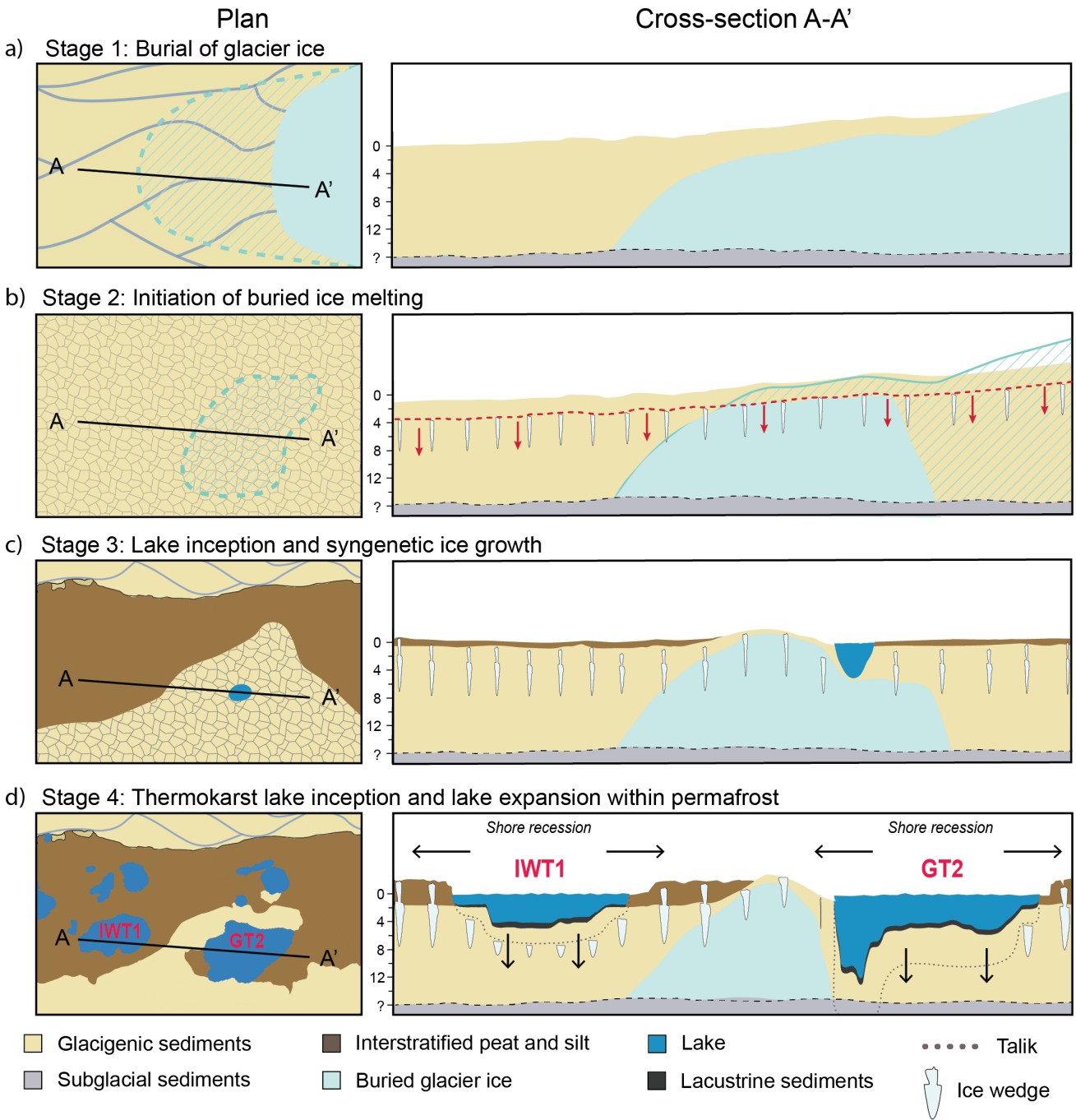

**Figure 7: Schematic diagram showing the sequence of formation of lakes in terrain underlain by relict glacier ice.**