# Peer review of "Contrasted geomorphological and limnological properties of thermokarst lakes formed in buried glacier ice and ice-wedge polygon terrain"

_The Cryosphere, 2021_

## Author Comment (AC1)

**Responses to the Reviewer's comments**

Comments by the two referees were very enlightening and their suggestions useful; we are grateful for their input. His/her careful reading of the manuscript and his/her good knowledge of the subject-matter allowed providing relevant suggestions and additions to the manuscript. We treat each point raised in detail and with great interest.

Note that the line numbers given in this response refer to the revised version of the manuscript in track changes mode.

**Referee #1**

Major edits

Comment 1:
**Referee #1: The Results section was confusing in terms of structure, order of the data, how the data were discussed, etc. Some examples of confusing groupings, structures, and statements:**
Authors: We agree that the Results section needs improvements in terms of structure. First, we changed the names of the subsections so it matches the structure of the Methods section to facilitate reading and understanding. We also separated the 'lake sediments' and 'vertical structure of water column of lakes' into two distinct sections in the 'Results'. If the referee wants to offer more specific comments or guidance on how to improve this section, we would be more than happy to implement those recommendations.

**(a) Radiocarbon dates and glacier retreat timing are discussed in the "Spatial distribution" section.**
Authors: The purpose of the section is to show that lakes have formed in front of glaciers C-79 and C-93 as theses glaciers retreated since the Little Ice Age. The radiocarbon date originates from a thrust plate forming part of the Neoglacial moraine of glacier C-79, indicating the timing of the last major glacier advance. We modified this paragraph to move the focus on the lakes rather than the retreat of these glaciers.

**(b) In the "Morphology of lakes" the authors discuss deep vs. shallow, whether lake bottoms are smooth, etc., but the information is vague and doesn't seem to match the data in the figures. They say deeper lakes have smoother lake bottoms, but it doesn't look like they do from the figures. Is this statement based on statistical analyses?**
Authors: Here, we refer to the microtopography of the lake floor. In shallow lakes, the irregular lake floors were inherited from the ice-wedge polygons which are now submerged and degrading. We added a few details to clarify this: "*The GPR profiles indicated that these deeper lakes usually have smoother microtopography at the lake bottoms, whereas lake L also exhibits an irregular lake floor micromorphology in the shallowest areas (Fig. S2), reflecting the pattern of submerged polygon-patterned ground and degraded ice wedge under frost crack troughs (Bouchard et al., 2020).*"

We also added this sentence for the shallow lakes: *"The lake floor is irregular at a small-scale (microtopography), which is attributed to submerged polygons (see the video supplement in Bouchard et al., 2020)."*

**I think the addition of more data tables would help clarify the salient results, especially for the three lakes (G, K, L) that were analyzed in detail. I do not know how the written part of Results section can be made more organized and clear, but I ask that the authors rethink the structure of the Results section and try to reorganize if possible.**
Authors: In the supplementary materials, there is a data table (S3) that summarizes the lake morphology for all lakes, including lakes G, K, L. We moved this table with the main figures and tables to address the reviewer's suggestion.

Comment 2:
**Referee #1: P15 L24-27 Here you discuss that the two sediment cores reveal two distinct basin types, but given that you only have two lake cores, this statement is a reach. If you took cores from four lakes (two shallow and two deep) you might end up with four distinct basin types or one of the deep lake cores might look similar to one of the shallow lake cores. I think the number of samples (n=2) prevents you from definitively stating that there are two distinct basin stratigraphies for the shallow vs. deep lakes. Sample duplication is needed to make that general statement.**
Authors: We agree and added a sentence to clarify that more work is needed to make that general statement: *"However, more work is needed to establish a strong link between the origin of lakes and their sedimentation histories by collecting more sediment cores from different shallow and deep lakes in the Qarlikturvik Valley, but also in other glacierized arctic tundra settings."*

Comment 3:
**Referee #1: P22 Fig.1 - What data did you base the mapped "glacier positions" on? Need to support these glacier positions with field and/or remote-based data that definitively shows past glacier positions (moraines, ice-contact lake deposits, etc.).**
Authors: We added more information in the 'Materials and methods' section to clarify how we mapped the glacier positions: *"We used remote-based data to map glacier frontal positions of glacier C-79 and C-93 to investigate the formation of new lakes in the valley at the termini of these glaciers over the past 60 years: 1) historical aerial photos (1961, 1982; National Air Photo Library) 2) GeoEye satellite imagery (2010, pixel = 0.5 m); 3) Sentinel-2 (2016, 2020, pixel = 10 m) and 4) field measurements using a real-time kinematic (RTK; July 2011;Trimble R8). The positions refer to the contact between the ice and moraine material."*

Comment 4:
**Referee #1: The article would be strengthened by the inclusion of a data table showing the lake morphology results for all of the lakes.**
Authors: There is a data table in the 'Supplementary information' that summarizes the lake morphology for all lakes. We moved this table with the main figures and tables to address the reviewer's suggestion.

Minor edits

**Referee #1: P1 L20 – "remotely-sensed"**
Authors: Modification made.

**Referee #1: P1 L23 – Maybe change to "subsidence. They later"**
Authors: Modification made.

**Referee #1: P1 L27 – I do not understand what "if any" means here.**
Authors: We added "if there is still some remaining'' to clarify the sentence.

**Referee #1: P2 L3-5 – "massive ice, they"**
Authors: We modified the sentence to make it clearer: "*These ice-rich permafrost landscapes are experiencing thermokarst, through the thawing of near-surface ice-rich permafrost or the melting of massive ice, which may result in land subsidence and ponding.*"

**Referee #1: P2 L6-8 – long sentence**
Authors: We agree. The sentence was divided into two shorter sentences: "*In flat-lying terrains, thermokarst processes often result in the formation of numerous wetlands, ponds and lakes. This creates or modifies existing 'limnoscapes' (lake-rich landscapes) through thermal and mechanical erosional processes as well as thaw consolidation and subsidence beneath waterbodies.*"

**Referee #1: P2 25-26 – This sentence is confusing.**
Authors: We agree and changed it to: *"Some of these landscapes are now experiencing climate-driven renewed deglaciation leading to a second phase of post-glacial landscape evolution."*

**Referee #1: P2 L31 – em dash in "2–5"**
Authors: Modification made.

**Referee #1: P3 L7 – "amounts'**
Authors: Modification made.

**Referee #1: P4 L6 – em dash in "2–3"**
Authors: Modification made.

**Referee #1: P4 L15 – em dash in "1981–2020"**
Authors: Modification made.

**Referee #1: P4 L15 – should maybe be "189 mm/yr" or "189 mm yr-1"**
Authors: Modification made.

**Referee #1: P4 L17 – em dash in "100–500"**
Authors: Modification made.

**Referee #1: P6 L23 – "lakes"**
Authors: The sentence has been changed to address a previous comment (structure of the results section), so it should now be singular.

**Referee #1: P7 L9-11 – This is a run-on sentence.**
Authors: We divided this sentence into smaller sentences: "*We also profiled the same two lakes (G, K) as well as lake L in late winter under the ice cover (early June 2015) and during the ice-free period (July and August). The objective was to examine differences in water temperature and dissolved oxygen (DO) between lake types, and discuss the effects of water depth and vertical structure on GHG emissions and fish habitats.*"

**Referee #1: P7 L11 – "discuss" seems like a strange verb here.**
Authors: We replaced it by "address".

**Referee #1: P7 L21 – "Patterns of distribution emerge on this level" is confusing. On what level?**
Authors: We replaced it by "Patterns of distribution emerge in the valley with higher densities, […]"

**Referee #1: P7 L24 – Which valley?**
Authors: We added "Qarlikturvik".

**Referee #1: P7 L25 – "clustering in short distance" is vague.**
Authors: The "r < 0.85 km" in parenthesis specify the distance.

**Referee #1: P7 L28-30 – Here you say "highest densities occur directly in front of contemporary glaciers and within the extent of local mountain glaciations," but based on Fig. 6d it looks like there are high densities within the mapped extent of the LIS too. Please clarify.**
Authors: We agree. The purpose of figure 6d is to show that highest densities of lakes are typically found in association with the former positions of local glaciations, but also with the LIS. We forgot to mention it in the text, so we added "and the LIS" to the sentence.

**Referee #1: P7 L30 – Need "yr" in "120 14C yr BP"**
Authors: Modification made.

**Referee #1: P8 L13 – "bottoms"**
Authors: Modification made.

**Referee #1: P8 L16 – "K and L are 5.5 and 5 m, respectively, below"**
Authors: Modification made.

**Referee #1: P8 L18 – please specify again what the "second group of lakes" are – the shallow lakes?**
Authors: We added "shallow lakes" in parentheses.

**Referee #1: P8 L22 – "platforms"**
Authors: Modification made.

**Referee #1: P8 L26 – "depths"**
Authors: Modification made.

**Referee #1: P9 L7 – Please specify which "unit" you are talking about here. "Did not reach the bottom of this unit" is confusing due to vagueness.**
Authors: We added "the bottom of unit A" to clarify this sentence.

**Referee #1: P9 L17 – Please specify how moisture content is measured/reported here. Is it gravimetric or volumetric?**
Authors: We added "gravimetric water content" to clarify it was measured. (P8 L27)

**Referee #1: P9 L28 – Do you mean "shifts" instead of "sifts"?**
Authors: Yes, we made the correction. (P9 L31)

**Referee #1: P9 L32 – what does "bottomed" mean? Do you mean "bottom"?**
Authors: Yes, we made the correction.

**Referee #1: P10 L13 – "kicked saturation level up again" is too informal. Maybe "increased saturations levels"**
Authors: We changed it to "increased".

**Referee #1: P10 L13-14 – "(unpubl. data)." Why not just publish the data here? Is the plan to publish it in an upcoming paper – therefore it cannot be published here? Is the author of those data not an author on this paper?**
Authors: These data will be published in an upcoming paper. The author of this data is Isabelle Laurion, and she is one of the coauthor of this paper.

**Referee #1: P10 L15 –** Maybe say "On its margin" rather than "On its side"
Authors: We agree and made the modification.

**Referee #1: P11 L13 – is there a better way to say "broken horizon"? Disturbed? Faulted?**
Authors: We agree and changed it to "disturbed horizons"

**Referee #1: P11 L32 – I think there is an extra space between "content" and "measured"**
Authors: We removed the extra space.

**Referee #1: P11 L32 – Please specify which unit you are talking about here.**
Authors: This sentence has been changed to address another comment.

**Referee #1: P12 L12 – "which do not reflect"**
Authors: Modification made.

**Referee #1: P12 L13 – "similar to"**
Authors: Modification made.

**Referee #1: P11 L14 – delete "down" after the word "settle"**
Authors: Modification made.

**Referee #1: P12 L15 – should be a comma after "terrestrial runoff,"**
Authors: Modification made.

**Referee #1: P12 L15 – Maybe change to "aeolian activity, and/or 3)"**
Authors: We agree and made the modification.

**Referee #1: P13 L8 – Capitalize "Last Glacial Maximum"**
Authors: Modification made.

**Referee #1: P13 L9 – Maybe replace "in" with "through" so it reads "channeled through major"**
Authors: Modification made.

**Referee #1: P13 L9 – "terminated"**
Authors: Modification made.

**Referee #1: P13 L13 – delete dash between "and" and "meltwater"**
Authors: Modification made.

**Referee #1: P14 L8-10 – Can you be more specific than "since the mid-Holocene" here in terms of timing?**
Authors: We changed it to"~7-8 ka" to reflect the results presented in the references.

**Referee #1: P14 L10-12 – This sentence "This situation applied ..." is confusing. Do not understand what you mean.**
Authors: We changed it to: "*These warmer periods likely initiated or accelerated ice melt when (1) a thin layer of sediments covered the ice, or (2) topographic depressions allowed the accumulation of snow and water, hence overall warmer conditions, which further accelerated the melt.*"

**Referee #1: P14 L18 – "preceded"**
Authors: Modification made.

**Referee #1: P14 L23 – I don't think it should be a semi-colon in "year; the rate" because the second clause is not a complete sentence.**
Authors: We agree and we divided this sentence into two shorter sentences: "*Once a lake gets deeper than the maximum thickness of the winter ice cover (~2 m ± 20 cm in the valley as measured by Prėskienis et al., 2021), it will continue to grow laterally (thermo-mechanical erosion) and vertically (subsidence) by thermokarst processes each year. The rate of expansion depends on local climatic conditions, ground-ice content and lake bed temperature (Fig.7d).*"

**Referee #1: P14 L28 – Put period after the citations.**
Authors: Modification made.

**Referee #1: P13 L31-33 – Can you provide a citation for the statement that kettle lakes should be very smooth and close to a perfect circle? There are kettles, like Walden Pond in Massachusetts, that are not circles and that do not have smooth shorelines. So, the statement needs a citation.**
Authors: We agree that this statement needed a citation. In the literature, kettle holes and lakes are typically described as roughly circular, steep-walled or inversed-conical However, to our knowledge, more precise descriptions of the morphology of these glacial features remains very rare. As a result, we conducted additional analyses to further investigate the morphology of glacial lakes formed in proglacial outwash deposits in front on glaciers C-79, C-93 and C-67, which is adjacent to C-79. A WorldView image (2010, pixel = 0.5 m) of the glacier termini served as the basis for mapping the glacial lakes (n=400) and investigate their morphology. We obtained an average shoreline development index of $1.1 \pm 0.1$ and average elongation index of $1.6 \pm 0.1$, indicating that these recent glacial lakes are slightly oval rather than truly circular. The more oval and elongated lakes are aligned along the inter-morainal swales, which likely influence their shapes. These data will be published in the open-access Nordicana D data repository at Centre d'études nordiques (CEN – Centre for Northern Studies) (http://www.cen.ulaval.ca/nordicanad/) with the other datasets.

We added more details in the Methods, Results and Discussion sections to support the statement:
- Methods: *"For comparison, the morphological attributes of glacial lakes (i.e. kettle lakes) formed in proglacial outwash deposits in front on glaciers C-79, C-93 and C-67 were also calculated. Very few studies have studied kettle lake morphology, but these lakes are typically described as enclosed and steep-sided depressions, roughly circular and inverse-conical (Fay, 2002; Gorokhovich et al., 2009; Borsellino et al., 2017)"*
- Results: *"In addition, glacial lakes (n=490) located near the front of glaciers C-79, C-93 and C-67 have an average shoreline development index of $1.1 \pm 0.1$ and average elongation ratio of $1.6 \pm 0.1$, indicating the shorelines are relatively regular and are mostly oval-shaped."*
- Discussion: *"The shorelines of glacial lakes, such as kettle lakes, are expected to be very smooth and roughly circular or slightly oval-shaped, as shown by the morphological analysis of glacial lakes located at the termini of glaciers C-93, C-79 and C-67. However, most lakes studied here displayed slightly irregular shorelines (Fig. S3)."*

**Referee #1: P15 L5 – "of evaporation and/or partial or complete"**
Authors: Modification made.

**Referee #1: P15 L11 – "It influences the mixing"**
Authors: Modification made.

**Referee #1: P15 L23 – Since all other verbs are past tense, change to "controlled"**
Authors: Modification made.

**Referee #1: P15 L24 – Looks like there is a high dash between "L" and "The stronger."
Please fix.**
Authors: Modification made.

**Referee #1: P16 L1 – "the open-water period as compared"**
Authors: Modification made.

**Referee #1: P16 L17 – This part is confusing since you say "There are four key arguments"
but then you only list three. Is the "Moreover, analysis of lake morphometry" section the
fourth argument? If so, it needs to be numbered and indented.**
Authors: We agree and changed it to "*There are three arguments […]*".

**Referee #1: P16 L31 – Replace "It suggests" with "This suggests"**
Authors: Modification made.

**Referee #1: P17 L4 – "and that slumping and thaw subsidence"**
Authors: This addition will change the sense of the sentence.

**Referee #1: P17 L6 – "hydrological and ecological"**
Authors: Modification made.

**Referee #1: P24 Fig. 1 – In the left part of the figure the study area seems to be shaded in
(light yellow). Please mention this in the caption or delete the shading. As is, it is confusing**
Authors: We agree and added this sentence in the caption: "*The yellow shaded area shows the
southwestern plain of Bylot Island*".

**Referee #1: P28 L3 (Caption Fig. 4) – "slump found as buried glacier ice" is confusing.**
Authors: We removed "found" and added commas to make it clearer: *"We interpreted the
massive ground-ice, exposed at the headwall of thaw slump, as buried glacier ice on the basis of
cryostratigraphic, crystallographic and geochemical analyses"*

**Referee #1: P28 Fig.4 – On both Lakes (K and G) the second panel in black does not have a
label. Please label.**
Authors: We added labels to all panels.

**Referee #1: P30 Fig.6 – Since early June is actually spring, I think this should be labeled as
"late spring" and not late winter.**
Authors: We suggest replacing 'late winter' and 'summer' by the months during which the
measurements were taken. Initially, we used 'late winter' to better reflect the field conditions
when snow and lake ice has not started to melt or has just started. We also replaced 'late winter'
by early June in the "Results" section and the caption.

**Referee #1: P25 Fig.2b – Please provide a color scale and key for this figure. Even though the caption states that red denotes sediment accumulation, the other colors are not explained and the scale of the colors is not explained.**
Authors: We added more details in the caption: *"Sediment accumulation at the glacier front, which are drier and un-vegetated areas, are represented by red colours on the images. Wetter areas are shown in blue, and vegetated areas are distinguished by teal and yellow colours."*

**Referee #1: P25 L2 – "120 yr BP to present" or "~1900 to 2020"**
Authors: We changed it to "120 yr to present".

**Referee #1: P25 L7 – "unvegetated"**
Authors: Modification made.

**Referee #1: P33 A2 – In caption, please explain the open circles and the error bars. Are the error bars based on a 2-sigma or 95%? What are the open circles denoting – are these outliers? Total ranges?**
Authors: We added details in the caption: *"The thick line marks the median value. The bottom and the top of the box correspond to the first and third quartiles, respectively. The whiskers show the range of observed values that are not within the first and third quartile but not further away than 1.5 times the interquartile range (IQR) from the hinges, and open circles represent outliers."*

**Referee #1: P31 Table A4 – Why are the thaw settlement values red and bold?**
Authors: We removed this table (see next comment).

**Referee #1: P31 Table A4 – Please provide a citation for footnote 3 "Values typically found in these materials."**
Authors: We agree that a citation is required to support this table. However, we decided not to provide thaw settlement values due to a lack of data available., laboratory measurements of the soil porosity and volumetric or volumetric ice content is required to accurately estimate the potential thaw settlement of the two uppermost units (peat and silt, glaciofluvial sands). As a result, we removed this sentence: *"The highest estimates for potential thaw settlement varied from 1.2 m in gravel and sandy soils to 2.4 m in very ice-rich silty and peaty soils"*, and we replaced it by:

*"The cryostratigraphic context of the Qarlikturvik valley is not conducive to the formation of deep depressions. The uppermost unit consists of interstratified peat and silt (thickness ~ 2–4 m) with high volumetric ice content (74.6 ± 10.6 % ;Veillette et al., unpublished data), while the underlying unit is glaciofluvial sand and gravel, which typically have low ice contents and minimal expected settlement. Given the thickness of the silt and peat sequence and low ice content in the glaciofluvial sands, the amount of intrasedimental ice is not sufficient to create lakes with depths reaching up to 12 m, even if all the intrasedimental ice melted and drained out of the soil porosity."*

---

## Author Comment (AC2)

**Responses to the Reviewer's comments**

Dear Dr. Kokelj,

Thank you for the feedbacks on our manuscript submitted for publication in *The Cryosphere*. We greatly appreciated the comments and we have made substantial changes to the manuscript. Below are point-by-point responses to all comments and questions.

Note that the line numbers given in this response refer to the revised version of the manuscript in track changes mode.

**General comments**

Comment 1:
**Referee #2:**
**Overall the paper is fairly well-written, however, some sections require editorial clarifications. I think that improvements can be made in how the study is framed. The introduction focuses primarily on glaciated permafrost environments that host relict ice. There is a brief mention of thermokarst lake development in areas with wedges and segregated ice. While the authors emphasize that the paper is focused on lakes developing by delayed melting of buried glacier ice, a significant portion of the data presented in this manuscript seems to pertain to shallow thermokarst lakes that have developed due to the thawing of wedge ice. Even the conceptual model presented as a summary nicely illustrates two thermokarst lake types, but curiously, discusses the origins of only one. I think that the Authors have a great opportunity to cast the paper as a contrast between thermokarst lakes of varying origins, highlighting differences in their physical and limnological conditions and future sensitivity to change. This broadening of the focus would make a more compelling paper where issues like heterogeneity in lake type and physical and limnological conditions could be nicely linked to variation in the geological/permafrost/geomorphic history of the landscape. If the Authors chose not to frame the paper more broadly as suggested here, then they should remove the data and descriptions of lakes that have not developed in settings hosting relict ground ice because the material doesn't fit well with how the paper is currently scoped. This later adjustment would create a more focused paper consistent with its title and objectives.**

Authors:
- We agree with the referee's suggestion to frame the paper more broadly. We modified the manuscript to better demonstrate the contrast between thermokarst lakes of varying origins. The changes made are detailed below in the other general and specific comments.
- We also change the title to better reflect the comparison between the two lake types.
- To enhance this contrast, we added results from the diatom analysis.
- As for the conceptual model, Bouchard et al. (2020) developed a conceptual model for lake IWT1, which developed due to the thawing of ice wedge. However, we added a short paragraph that summarizes the different stages of development, while referring to the original article by Bouchard et al. (2020) for more details.

*"In a previous study, Bouchard et al. (2020) presented a four-stage conceptual model for lake IWT1 (named Gull Lake) that describes thermokarst inception and evolution in syngenetic ice-wedge polygon terrain during the Holocene. Based on this model, lake IWT1 developed in a pre-existing topographic depression (~1-2 m) that collected snow and meltwater (stage 0, initial conditions). The first phase of thermokarst started at around 2100 BP in response to active layer deepening and ice wedge melting, which initiated the development of small and shallow ponds over the degrading ice wedges (stage 1). Thermokarst ponds started to coalesce with neighbouring water bodies over and at the edge of ice-wedge polygons to form a small lake (stage 2). Over time, this lake expanded in the ice-rich polygon terrace because of surface permafrost degradation via lateral thermal erosion and vertical thaw settlement and consolidation in the ice-rich silt-peat terrace, and eventually in the underlying glaciofluvial sediments (stage 3). The last stage suggests a possible long-term future scenario where the lake disappears through the gradual gyttja accumulation and lake infilling or lake drainage, which can sometimes be catastrophic (Bouchard et al., 2020 and citations therein). The conversion of these aquatic ecosystems to terrestrial or wetland ecosystems is usually followed by a reactivation of old ice wedge networks or growth of cryogenic mounds as permafrost aggrades in unfrozen drained lake deposits once exposed to cold temperatures, which eventually begin a new phase of the thaw-lake cycle (Mackay and Burn, 2002; Jorgenson and Shur, 2007).*

Comment 2:

**Referee #2: Referencing past or related work could be slightly improved. Several of the concepts or ideas that the Authors have assessed with focused field investigations build on or relate to work from other regions, that may have been conducted at broader spatial scales, or pertain to concepts that have been integrated into ground ice modeling approaches. I think that making it clear that climate sensitivity of permafrost preserved glaciated is an established concept that this study builds on is important to clarify because it would better highlight that the paper findings are relevant to lake types and environments found across large parts of NW Canada, Alaska, and Siberia. I would note that there is also a fair bit of limnological and paleoenvironmental work on lakes developed in ice-rich glaciated terrain, so making better connections with some of that work from the western Canadian Arctic and Alaska could help draw interest from a broader Arctic change science community.**

Authors: In the introduction, we added a few references (in blue) to relate to work from other regions. Together, these recent citations report permafrost vulnerability from different regions, including ice-cored terrain.

- *"Arctic landscapes underlain by ice-rich permafrost are highly vulnerable to climate change and permafrost degradation (Segal et al., 2016; Rudy et al., 2017; Lewkowicz and Way, 2019; Kokelj et al., 2017; Nitzbon et al., 2020, Douglas et al., 2021).*

- *"These ice-rich permafrost landscapes are experiencing thermokarst, through the thawing of near-surface ice-rich permafrost and/or the melting of ice wedges or massive ice, which may result in land subsidence and ponding (Kokelj and Jorgenson, 2013; Farquharson et al., 2019; Liljedahl et al., 2016; Edwards et al., 2016; Jorgenson and*

*Osterkamp, 2005)*.”
Note: The last citation is less recent, but the authors specifically refers to glacial thermokarst of ice-cored moraines in Alaska. We did not include citations referring to rapid mass movements such as thaw slumps.

- “*The formation and growth of these lacustrine ecosystems have important effects on the hydrology, ecology, biogeochemistry and geomorphology of affected landscapes (*Vonk et al., 2015).”

- “*The persistence of thick beds of buried Pleistocene glacier ice in contemporary permafrost environments indicates that deglaciation is still incomplete (Astakhov and Isayeva, 1988; Everest and Bradwell, 2003; Kaplanskaya and Tarnogradskiy, 1986; Lenz et al., 2013).*”

Comment 3:
**Referee #2: Returning to the issue of how the paper is framed, I would encourage the Authors to further develop the ideas that permafrost/ground ice/Quaternary history can lead to significant contrasts in the physical and limnological conditions of Arctic lakes and ponds. Framing results in this geographical/geomorphic context could help the Authors make more focused, yet broadly applicable statements about variability in Arctic aquatic environments based on what has been learned in this study. I think that this is a more interesting and useful avenue for discussion (the spatial variability in lake conditions is related to ground ice/geological history) than echoing generalities about the sensitivity of Arctic aquatic systems.**

Authors: We agree and we mentioned this in the abstract, introduction and:

- Abstract (P1 21-23) : “*These remnants of glacier ice buried and preserved in the permafrost strongly contribute to the high spatial variability in ground ice condition of these landscapes, leading to the formation lakes with diverse origin, morphometric and limnological properties.*”
- Introduction (P3 L16-17): “*The Quaternary geology of the eastern Canadian Arctic records several glaciations by ice sheets and local mountain glaciers, which means that the landscape stores vast amounts of buried glacial ice, and there is potential for significant post-glacial landscape change associated with the ablation of this buried ice. The resulting landscape can be covered with a large number of thermoskarst lakes of diverse origin that impact their physical and limnological properties.*”
- Discussion (P12 L2-4): “*The ice-marginal permafrost environment in the Qarlikturvik Valley is highly heterogeneous, as ground ice types can vary and coexist over short distances, leading to significant small-scale differences in lake types, their physical and limnological conditions, as well as their vulnerability to climate drivers and disturbances.*”

Comment 4:

**Referee #2: There are one or two sections in the methods and results where the purpose of the analyses needs to be clarified. Specifically, it was not obvious to me how some of the spatial analyses were linked to specific research objectives. I think that minor editorial modifications could address this concern. Also, if lake types were dichotomized by origin or type then summarizing their morphological characteristics could be of interest. In this regard, providing a more-clear rationale for presenting data on different lake types would be useful.**

Authors: For the spatial analyses, we examined the spatial distribution and organization of lakes in order to investigate their spatial distribution in relation to former known glacier positions in the valley and broader southern plain. We wanted to investigate whether lakes are more clustered or abundant along past glacier margin positions, regardless of their origin.

- We added a third specific objective to better link the spatial analyses to the objectives presented in section 1 (Introduction, P3 L 28-30):
  *"The specific objectives were therefore (1) to compare their physical and limnological properties; (2) to examine the possible link between the spatial pattern of lakes and past glacier positions in the Qarlikturvik Valley and broader southern plain, and (3) to develop a conceptual model of lake inception and evolution, with a focus on lakes formed by the delayed melting of buried glacier ice."*

- We also added some details at the beginning of section 3.2 (P6 L 2-5), and we can provide more details if required:
  *"We examined the spatial distribution of lakes to examine possible association with past glaciers positions in the Qarlikturvik Valley and the broader southern plain on the island. This can provide additional evidence on the glacial origin of lakes because these ice-marginal zones are often comprise discrete bodies of glacier ice left behind by a retreating glacier and buried underneath sediment."*

  P6 16-17: *"A high spatial clustering suggests that the spatial distribution of lakes is dependent on an external variable which we interpreted as the probable presence of patches of buried glacier ice."*

Comment 5:

**Referee #2: Finally, the organization of the paper requires some improvement. Specifically, in the version of the Manuscript that I have reviewed many of the figures are cited in the wrong order. The Authors include figures, an appendix, and supplementary materials. I would suggest including the figures in the appendix in the main body of the manuscript or the supplement.**

Authors: We apologize for that, and we corrected the order of the figures and table the manuscript. We also moved the figures in the appendix to the supplementary material.

**Specific comments**

**Referee #2: P1 Abstract – Overall the abstract is well-written, but seems to omit aspects of the paper on thermokarst lakes in polygonal terrain on glaciofluvial plain. A significant portion of the data and figures in the paper contains information on the latter type of lake. So, strictly speaking, given the content of the paper, the abstract seems incomplete.**

Authors: We agree and we changed the abstract to include aspects of the paper on thermokarst lakes in polygonal terrain.

P1 L25-27: "*. Our results suggest that initiation of thermokarst lakes in the valley was either triggered from the melting of buried glacier ice or permafrost intrasedimental ice and ice wedges.*"

**Referee #2: P1 L18-21 – Minor modification to better contextualize the statement is required here. Are all relict bodies of ice going to melt, over what time scale, and with what projected warming?**
Authors: We added a few details in this sentence to answer these questions (P1 L18-21):
"*It is expected that large parts of glacier ice buried in the permafrost will melt in the near future, although the intensity and timing will depend on local terrain conditions and the magnitude and rate of future climate trends in different Arctic regions. The impact of these ice bodies on landscape evolution remains uncertain since the extent and volume of undisturbed relict glacier ice are still unknown.*"

**Referee #2: P2 L30-32 – It would be good to attribute the broad-scale linkage between "permafrost preserved-glacial landscapes and thermokarst vulnerability" in this sentence to the research that produced this conclusion.**
Authors: We added these two references "Segal et al., (2016), Kokelj et al., (2017)", which demonstrated the link between glaciated permafrost terrain and thermokarst vulnerability.
*"The broad distribution and the substantial amount of ground ice in glaciated permafrost landscapes make it highly vulnerable to disturbances, such as thermokarst, under the ongoing climate warming (Kokelj et al., 2017, Segal et al., 2016)"*

**Referee #2: P3 L21-30 – Overall, I find the introduction to be well-written. As indicated in my summary, after reading the manuscript I feel that the background, and specifically the objectives seem to frame only a portion of the data that is presented in the paper.**
Authors: We modified the objectives and hypothesis to better include all the data presented in this study:
"*Here, we investigate the inception and evolution of twenty-one lakes from the lower reach of a glacial valley on Bylot Island, which presents heterogeneous permafrost ground ice conditions. We hypothesized that thermokarst lakes have different origins and exhibit differences in their physical and limnological conditions as well as future sensitivity to change. In the Qarlikturvik Valley, remnants of buried glacier ice in lowlands slowly melted during the Holocene, which created deep depressions that formed glacial thermokarst lakes, while the thawing of an ice- and*

*organic-rich polygonal terrace created shallow thermokarst lakes. The specific objectives were therefore (1) to compare the physical and limnological properties of these two types of thermokarst lakes; (2) to examine the link between the spatial pattern of lakes and past glacier positions in the Qarlikturvik Valley and broader southern plain, and (3) to develop a conceptual model of lake inception and evolution, with a focus on lakes formed by the delayed melting of buried glacier ice."*

**Referee #2: P4 L6-8 – It would be helpful to show the Eclipse moraine on the maps in Fig. 1. After reading the manuscript, I wonder if the Eclipse moraine is that shown in Figure 6d. It would be useful to reference the figure at this point in the manuscript. Also, confirm on the figure the direction of ice flow of the LIS vs the alpine glaciers.**
Authors: We added the limit of the foreign drift left by the Eclipse Glaciation and direction of ice flow of the LIS in figure 1.

**Referee #2: P4 L12 – Spelling correction – change "through" to "trough".**
Authors: Modification made.

**Referee #2: P4 L13-16 – The Authors could add that more generally, the valley has a diversity of depositional environments, including glacial/ice-cored deposits of varying age, making it an excellent place to study ice types and thermokarst lake development under varying ground ice and terrain conditions.**
Authors: We agree with the suggestion, and we suggest to replace these two sentences "*This lake-rich valley is an ideal location to study buried glacier ice dynamics in thermokarst-affected and glaciated permafrost landscapes. With glaciers ending within the continuous permafrost zone, this typical glacial valley represents a small geosystem shaped by proglacial, paraglacial and periglacial processes*"

by:

"*With glaciers ending within the continuous permafrost zone, this lake-rich valley represents a typical glaciated valley geosystem that incorporates numerous depositional environments associated with ice-marginal, proglacial, paraglacial and periglacial processes, which makes it an ideal location to study ice types and thermokarst lake development under varying ground ice and terrain conditions.*"

**Referee #2: P4 L3 – If the net pattern on Figure 1 represents polygonal patterned ground this should be added to either the legend or the caption. The Authors should also indicate in the legend that the dotted green lines indicate "former glacier positions".**
Authors: We made the changes in the legend and caption.

**Referee #2: P4 L3 – Remove the word "an"**
Authors: Modification made.

**Referee #2: P4 L31-32 – The Authors cite measurements from Somerset and Devon Islands for permafrost thickness as a range from 100 to 500 m. Given the climate and geological history, where does the study area fit in this range?**
Authors: We changed it for estimations based on temperature measurements on the island conducted by Dr. Moorman.
"*Permafrost thickness was estimated to be at least 200–400 m based on shallow ground temperature measurements on the island (Moorman, 2003)*."

**Referee #2: Materials and Methods – Here it would be useful to mention that various spatial scales were used to investigate the contrasting roles of buried and wedge ice in the formation and evolution of thermokarst lakes.**
Authors: At the beginning of section 3 (Materials and methods; P5 L4), we already mention:
"*Two spatial scales were used to investigate the role of buried glacier ice […]*".

**Referee #2: P6 Section 3.2 – By examining the nature of lakes concerning different glacial limits is there a time factor that is sampled across that the Authors should/could consider in their analysis?**
Authors: Depending on local terrain stability and conditions, including thickness and/or nature of the sediment cover, a lake formed recently could be associated with an older glacier position. There is no indication that a lake formed in older glacial sediments should be older than other lakes that developed in younger deposits.

**Regarding the clustering analyses to investigate lake distribution: it would be useful to better link the method and results to a specific research question. Also, it is stated that the analyses assume that objects, which I gather are lake centroids, can be distributed anywhere in the region of interest. Is this assumption valid if the input data is lake centroid, but the lake is some area greater than the lake centroid? The centroid of an adjacent lake could not be located within another lake, and therefore could not be distributed anywhere on the spatial surface.**

**Regardless, a bit more detail on the analyses, and later on an explanation of the results would be helpful.**
Authors: As mentioned earlier, we added a few details to better link the method and result (P5 L 27-30 + P6 L4-12). We also added a specific objective at the end of the introduction.

**Referee #2: P5 Section 3.3– On what basis were the 21 lakes selected? It seems that a portion of the lakes was not associated with moraine deposits. How does this sampling design fit with the broader study objectives?**
Authors: These lakes are the largest and deepest lakes in the valley, while most of the other smaller waterbodies are very shallow thaw ponds. Despite the absence of continuous glacial deposits, it should be noted that glacio-fluvial outwash plain once occupied the entire valley floor, which contributed to the burial of glacier ice. Earlier glacial landforms/deposits were likely modified or buried beneath outwash deposits. Also, most of these lakes are aligned with past glacier positions in the valley as shown by mounds of ice-contact deposits.

We added a few details to briefly explain the main reason for choosing these lakes (P5 L8-9): "*The studied lakes are among the largest in the valley and most of them are close to former glacier positions.*"

**Referee #2: P7 L13 – Section numbering is incorrect. This should be 3.4 Lake sediments ....**
Authors: Correction made.

**Referee #2: P6 L9 – The Authors identify 3 lakes (G, K, and L). Why were these particular lakes selected for study? Given the topic of the paper, I had assumed that these were lakes that had developed due to the degradation of relict glacier ice, but they seem to have been selected to contrast conditions in different lake types. It would be helpful if they were labeled more intuitively, perhaps by the type of physiographic environment that they occur in. Further to earlier points in this review, if the paper is only focused on "Thermokarst lakes formed in buried ice" it is odd to include lake G which I believe has a contrasting origin.**

**To better clarify the nature of data collection I suggest adding "the stratigraphic profiles of lake bottom sediments, and water column profiles of temperature and dissolved oxygen."**

Authors:
- We changed the labels of the lakes to GT 1 to 8 (**G**lacial **T**hermokarst) and IWT 1 to 13 (**I**ce **W**edge **T**hermokarst).
- This section has been divided into two separate sections in response to a comment made by the first referee. As a result, we only added part of the suggestion: "*We selected two nearby lakes (IWT1 and GT1) exhibiting different morphometry to compare the stratigraphic profiles of lake bottom sediment.*"
- We also modified the title of sections 4.3 and 4.4 as suggested.

**Referee #2: P7 L15 – Remove the word "lake" to read "...lakes K and L are the deepest in the valley"**
Authors: Modification made.

**Referee #2: P7 L16-18– Slight editorial adjustment is suggested to reduce redundancy. "...... degraded ice- wedge troughs confirmed that the shallower lake (G) is evolving through the ....."**
Authors: Modification made.

**Referee #2: P8 L5-12 – This section of the methods was surprising because it was not clear in the introduction that contrasting limnological conditions between lake types were a study objective. As the paper is currently scoped, these methods do not seem relevant to the study.**
Authors: As suggested earlier in the general comment, we have decided to present the contrast between thermokarst lakes of varying origins. In doing so, these methods are now relevant to the study.

**Referee #2: P7 – The figures appear to be out of order, as Fig. 6c and d are referred to before Figure 6a and b, or Figure 5, and so forth.**
Authors: We corrected the order of the figures, and we changed the order of the panels in figure 6.

**The analyses of lake and pond conditions would benefit from showing a frequency distribution of waterbody occurrence by size. Consider presenting these results for the entire population and then for the respective clusters identified in the analysis. Given the small size of many of the water bodies, how does this affect the clustering? Is the size of the lake associated with lake origin?**
Authors: We do not know for sure the origin of all lakes located in the southern plain, and so we don't think, at this stage, that we can associate the size of lake with a specific origin. Also, we did check this relation for the 21 studied lakes in the valley. Assuming that the deeper lakes have a glacial origin, we tested the difference between the two groups of lakes (glacial vs non-glacial) in terms of their size (area, maximum length) , or the correlation between shoreline morphology variables and basin morphometry, which was unconclusive. However, the results could be different with a larger number of lakes.

P7 L10-12: "*Correlation between shoreline morphology variables and basin morphometry (maximum depth) were tested using the non-parametric Kendall tau rank correlation for non-normally distributed data.*"

**Referee #2: P8 L20-28 – The purpose of this analysis and the meaning of these results is unclear. I am not sure what research question this analysis is investigating. Some additional narrative in the methods or here would help to clarify this point.**
Authors: This analysis can provide additional evidence on the glacial origin of lakes by their close association to past glacier margins. While highest density of lakes along past margins can suggest a glacial origin, the magnitude of clustering could reveal patterns caused by the occurrence of buried glacier ice masses. With glacial deposits being widespread in the island, clusters could suggest it could be dependant of an external variable, such as patches of buried glacier.

- As mentioned earlier, we added a few details in the Methods section (section 3.2)
- In Discussion (P13 L9-12):
  - "*We found that, even after accounting for landscape heterogeneity (i.e. high slope gradients, bedrock exposures), the lakes are still far more clustered when compared to a random spatial distribution. As a result, we propose that the clustering reveal patterns caused by the presence of patches of buried glacier ice. This provides additional evidence for supporting the glacial origin of these lakes.*"

**Referee #2: P8 L28 to P9 L2 – Can the Authors provide a bit more explanation of the TC figure 6b? For example, why is the active burial of glacier ice indicated by the red colors? What do the other red/orange areas in the floodplain of the braided stream valley indicate? Do the TC analyses indicate any evidence of pond development or expansion of water bodies as per Fraser et al., (2014)?**

Authors:

- We answered a similar question asked by the first referee. The colors are explained in the caption of the figure: "*The accumulation and movement of sediments in the outwash plain and at the glacier front are represented by red and orange colours on the images (dry and unvegetated areas; TC brightness). Wetter areas, such as eroding cliff or lake shore, are shown in blue representing increasing (TC wetness). Vegetated areas are distinguished by teal and yellow colours (TC greenness).*"

- We also added a few details in the ''Material and methods" section. Since this is minor component of the study, we think that details on the methods can be accessed through the references listed in the manuscript (P5 L19-23):
  "*We also used the Google Earth Engine Timelapse dataset (2000-2019) to visually assess terrain change and sediment accumulation at the glacier terminus based on Tasseled cap (TC) trend analysis of a Landsat image stack (Fraser et al., 2012; Gorelick et al., 2017; Nitze and Grosse, 2016). The tasseled cap transformation reduces the Landsat reflectance bands to three orthogonal indices called brightness, greenness and wetness (Crist and Cicone, 1984).*"

- Looking at Landsat reflectance trends with tasseled-cap indices (figure 2b), it shows a strong magnitude in the TC-brightness (color = red) trends at both the glacier margins, indicating dry and unvegetated surfaces. This corresponds to sediment accumulation onto the glacier surface, and represents a modern analogue of the burial of glacier ice. The brightness trend correlates well with the active burial of ice observed at the margins of glaciers C-93 and C-79m which is shown in the supplementary material (figure S5). We added a few details in section 5.2 (stage 1; P15 L14-18):
  "*The burial of glacial ice can still be seen today at the margins of many glaciers on Bylot Island. The TC brightness index exhibits a strong positive trend at the glacier margins, indicating dryer and unvegetated surfaces. This corresponds to sediment accumulation onto the glacier surface and represents a modern analogue of the burial of glacier ice. The brightness trend correlates well with the active burial of ice observed at numerous locations at the margins of glaciers C-93 and C-79 (figure S5).*"

- Red/orange areas in the floodplain of the braided stream valley:
  We added a few details in the caption to explain this (see above).

- We do observe evidence of pond development or expansion of water bodies as shown by an increase in values related to moisture/water. However, the lower resolution of Landsat images makes it difficult to fully assess the changes because of the size of lakes. Nonetheless, we added this sentence (P16 L26-28):
  "*The water sensitive index TC-wetness exhibits a moderate to strong positive trend for many lakes, driven by the gradual erosion lake shores containing ice-rich permafrost.*"

**Referee #2: Section 4.3 – It would be useful to lead off this section with a brief explanation as to why cores from these two particular lakes are presented.**
Authors: In section 3.4, we explain why we chose these two particular lakes, but we added some details (P7 L14-18):
"*We selected two nearby lakes (IWT1 and GT1) exhibiting different morphometry to compare the stratigraphic profiles of lake bottom sediment. According to the bathymetric surveys, lake GT1 is the deepest in the valley (max. depth = 12.2 m), and it lies directly next to an ice-contact deposits mound. We also sampled lake IWT1 (max. depth = 4.1 m) as lake bottom imagery revealed submerged ice-wedge polygons (~1 m depth) and degraded ice-wedge troughs, which confirmed that lake IWT1 is evolving through the melting of permafrost intrasedimental ice and ice wedges (see video supplement in Bouchard et al., 2020).*"

**Referee #2: Section 5 – This summary section identifies contrasts in characteristics of different lake types. There seems to be a great opportunity to elaborate this slightly to highlight the spatial heterogeneity in ground ice associated with ice-marginal environments leading to significant small-scale differences in lake types, their limnological conditions, and potential for thaw-driven change.**

Authors: We agree and we added this sentence as an opening statement to discuss both lake types (P12 L2-4): "*The ice-marginal permafrost environment in the Qarlikturvik Valley is highly heterogeneous, as ground ice conditions can vary and coexist over short distances, leading to significant small-scale differences in lake types, their physical and limnological conditions, as well as their vulnerability to climate drivers and disturbances.*"

**Referee #2: P13 L30-31 – Reference to Russian literature here is good, however, there are also examples from western Arctic Canada, where melt out of massive ice has yielded lakes with deep holes that are also prone to thaw slumping.**

Authors: This sentence refer specifically to stages of sedimentation in thermokarst glacial lakes, excluding lakes formed by glacial scoured or proglacial lakes. In replying to a comment made by the first reviewer, we added a reference for older "kettle" lakes studied in northern USA. Although there is literature on kettle/glacial/postglacial lakes in western Canada (e.g. Chapman Lake) or Alaska (e.g. Toolik lake), we did not find any studies reporting characteristics of lake bottom sediments and presenting similar stages of sedimentation for this type of lake. However, we would be happy to add any references that would be suggested to us.

**Referee #2: P12 – Given the data presented in the paper and the two lake types shown in the schematic, it would seem logical to frame the conceptual diagram here as a contrast of two types of thermokarst lake formation common in the study region (and in other ice-marginal permafrost preserved environments).**
Authors: Since the conceptual model for lakes that developed in syngenetic ice-wedge polygon terrain is already described in detail by Bouchard et al. (2020). We suggest presenting a summary of the developmental stages shown in this previous paper, and show the development of thermokarst lakes in buried glacier ice in more detail, which hasn't been done before. The differences in basin morphometries (i.e. depth) and sediment bottom stratigraphy allow to

strengthen the presumption that some lakes in the valley have been initiated by the melting of buried glacier ice, since the origin of lake G, for example, has been confirmed.

- We changed the section's name: "5.2 *Conceptual model of thermokarst lake development in polygonal terrain and buried glacier ice*"

**Stage 1. I presume glaciofluvial processes could also have eradicated buried ice as well. Is its preservation also possible in the glaciofluvial outwash plain sediments in the study region?**
Authors: Yes, its preservation is possible in the outwash plain.

**Consider adding some vertical scale or a stable reference marker to better visualize terrain evolution.**
Authors: We added a vertical scale.

**Stage 2-3. Do the ice wedges also get wider over time?**
Authors: That is correct. We made the change on the figure.

**Referee #2: P18 L21-24 – I think based on the strong links between lake characteristics and their geomorphic/ground ice environments that are shown in this study there is an opportunity for the Authors to develop more geographically focused comments on how results in this research contribute towards understanding variability in the thermokarst sensitivity of Arctic lakes.**
Authors: We added "*Spatial variability in ground ice conditions is an important factor driving lake inception, evolution and distribution on Bylot Island. This study confirms that glaciated permafrost terrain containing various types of ground ice, including buried glacier ice, can influence the spatial distribution of lakes, lake bathymetry, limnological properties as well as lake bottom morphology and sediment stratigraphy.*" However, perhaps this does not respond to the comment. We can make another change if the comment is clarified.

**Referee #2: P15 L26-27 – The authors state that their data shows a strong contrast between two lake types and then summarize conditions in one lake type. It would seem logical to summarize information on both lakes types since the data are presented in the paper.**
Authors: We agree and we summarized information on both lakes types in the first part of the conclusion.